# Metabolic control of progenitor cell propagation during *Drosophila* tracheal remodeling

Yue Li [1,2], Pengzhen Dong[1,2], Yang Yang[1], Tianyu Guo[1,2], Quanyi Zhao[3], Dan Miao[1], Huanle Li [1,2], Tianfeng Lu[1,2], Fanning Xia[1,2], Jialan Lyu[1,2], Jun Ma [2,4], Thomas B. Kornberg[5], Qiang Zhang [1✉] & Hai Huang [1,2✉]

Adult progenitor cells in the trachea of *Drosophila* larvae are activated and migrate out of niches when metamorphosis induces tracheal remodeling. Here we show that in response to metabolic deficiency in decaying tracheal branches, signaling by the insulin pathway controls the progenitor cells by regulating Yorkie (Yki)-dependent proliferation and migration. Yki, a transcription coactivator that is regulated by Hippo signaling, promotes transcriptional activation of cell cycle regulators and components of the extracellular matrix in tracheal progenitor cells. These findings reveal that regulation of Yki signaling by the insulin pathway governs proliferation and migration of tracheal progenitor cells, thereby identifying the regulatory mechanism by which metabolic depression drives progenitor cell activation and cell division that underlies tracheal remodeling.

---

[1] Department of Cell Biology, and Second Affiliated Hospital, Zhejiang University School of Medicine, Hangzhou, Zhejiang Province 310058, China. [2] Zhejiang Provincial Key Laboratory of Genetic & Developmental Disorders, Zhejiang University School of Medicine, Hangzhou 311121, China. [3] National Center for Cardiovascular Disease, Fuwai Hospital, 167 North Lishi Road, Xicheng District, Beijing 100037, China. [4] Institute of Genetics and Department of Genetics, Division of Human Reproduction and Developmental Genetics of the Women's Hospital, Zhejiang University School of Medicine, Hangzhou, Zhejiang Province 310058, China. [5] Cardiovascular Research Institute, University of California San Francisco, San Francisco, CA 94158, USA. ✉email: qiangzhang32@zju.edu.cn; haihuang@zju.edu.cn

In many contexts, adult stem cells reside in anatomical microenvironments––niches––where they remain quiescent until they are induced[1,2]. Activated stem cells can either self-renew or generate progeny that contribute to tissue maintenance and remodeling. There are many well-characterized examples. Injury induced by influenza infection or bleomycin exposure in the mouse lung induces quiescent epithelial stem cells to regenerate lung epithelium[3]. Intestinal stem cells accelerate the rate of cell division in response to tissue damage[4]. Bacterial infection increases stem cell proliferation and epithelial renewal[5]. Metabolic depression caused by energy and metabolic restriction, can also induce stem cells to remodel and reconstitute tissues, such as in the lungs of hibernating animals[6] or estivating frogs[7]. The present study investigates the mechanism by which metabolic deficits induce tissue remodeling. It examines the reconstitution of tracheal airway tubes that degenerate during metamorphosis in the fruit fly Drosophila. Previous studies identified tracheoblasts in the spiracular branches (SB) of the post-mitotic larval tracheal system[8] that are activated to replenish cell losses in decaying branches during metamorphosis[9]. Our findings show that these progenitor cells are activated by a process of metabolic deficit-induced signaling by the insulin and Hippo signaling pathways.

In Metazoans, insulin-signaling is a measured response to varied glucose and lipid energy levels that cells use to regulate metabolic activities. In the presence of insulin, the insulin receptor tyrosine kinase (IR) phosphorylates insulin receptor substrate proteins (IRS proteins), leading to the activation of the lipid kinase PI3K and membrane recruitment and activation of Akt[10,11]. A critical target of Akt is the evolutionarily conserved serine/threonine kinase 5′-adenosine monophosphate (AMP)-activated protein kinase (AMPK) that is inhibited by Akt-mediated phosphorylation[12]. Under energy-depletion conditions, phosphorylation and activation of AMPK are responses to increases in the cellular AMP/ATP ratio[13]. At the whole animal level, when glucose intake is reduced, the Drosophila insulin pathway is dampened and AMPK is activated[14].

The evolutionarily conserved Hippo signaling pathway[15] regulates cell proliferation and stem cell dynamics. The Hippo-Yes-associated protein (YAP) has an integral role in coupling nutrient-sensing, cellular proliferation, and differentiation[16,17]. The Ste20-like kinase Hippo (Hpo) phosphorylates and activates the NDR family kinase Warts (Wts), which in turn phosphorylates Yorkie (Yki), the Drosophila ortholog of the mammalian transcriptional coactivator YAP. Phosphorylation of Yki incapacitates its nuclear localization and association with Scalloped (Sd)/TEA domain (TEAD) DNA binding proteins in part through recruiting 14-3-3[18,19]. In contrast to the inhibitory regulation from the canonical Hippo signaling pathway, YAP also belongs to phosphorylation targets of a broad spectrum of kinases that can release the cytoplasmic retention of YAP and potentiate YAP transcriptional activity and target gene expression[20–22]. Yki, which is required for proliferation and tissue growth, promotes the transcription of target genes including cell-cycle and cell-death regulators[23,24]. In the Drosophila wing disc, genetic evidence suggests a direct interaction between Yki and transcription factors E2F1 and GAGA (GAF)[25,26]. Although both the insulin and Hippo signaling pathways have roles in controlling cell division, terminal differentiation, metabolism, and cell death, there is very limited evidence for cross-talk between these pathways. In cancer cell lines, AMPK- and Akt-mediated phosphorylation of YAP has been observed[27–30].

Here, we describe an interaction between these two pathways in controlling the proliferation and migration of tracheal progenitor cells. Our results show specifically that Akt activated by the insulin signaling directly leads to Yki phosphorylation. Our findings also reveal a molecular signature of tracheal progenitor cells, and identify targets of Yki-mediated transcription in the tracheal progenitor cells.

## Results

**Metabolic depression in pupal trachea.** The larval tracheal system is an interconnected network of branches including dorsal trunk (DT) tubes, dorsal branch (DB), transverse connective (TC), visceral branch (VB), spiracular branch (SB), lateral trunk (LT), ganglionic branches (GBs) in each of the 10 tracheal metameres (Tr1-Tr10; Fig. 1a,a', arrows)[31,32]. Clusters of tracheal progenitors are present in the 4th and 5th of the ten bilaterally symmetric SBs as shown in Fig. 1b, where red fluorescent protein (RFP) was expressed from a transgene containing a promoter fragment that labels tracheal progenitor cells[9]. To simultaneously image the DT in larval and pupal stages, we utilized a driver harboring the enhancer for delta that is preferentially expressed in the DT[33]. The progenitors in Tr4 and Tr5 migrated on respective TC branches until they reached DT during the third larval instar (L3) (Fig. 1a, b, arrowheads). Migrating progenitors then moved posteriorly on the DT at the onset of pupariation, wrapping around the DT and tracking its tortuosity (Fig. 1a', c–c'" and Supplementary Movie 1) The velocity of posterior movement was ~0.5 metamere/h (Fig. 1c–c'", Fig. 3l and Supplementary Movie 1). During pupariation, the Tr4 and Tr5 tracheoblasts replaced the metameres Tr6–Tr10 which are destined for destruction (Fig. 1a').

To investigate metabolic status during pupariation, glucose and ATP levels were assayed. The results showed a global decline of glucose abundance in white pupae, compared with that in L3 larvae (Fig. 1d). In Drosophila, the fat body is the functional homolog of mammalian liver and is the main reservoir for glucose, lipid and other nutrients[34]. To determine if glucose is allocated to trachea, glucose levels were measured in trachea isolated at several times during puparium formation. Tracheal glucose decreased during puparium formation, suggesting that the tracheal storage of glucose and its consumption were compromised at this stage (Fig. 1e). Glucose catabolism generates ATP, and as expected, the level of intracellular ATP in the trachea of white pupae (0 h APF) was lower than that in the trachea of L3 larvae (Fig. 1f). Together, these results show that energy levels were reduced during the larval–pupal transition and that tissue remodeling and morphogenesis inversely correlates with energy supply at this stage.

**Insulin and Yki signaling during the larval–pupal transition.** To investigate the genetic program that responds to metabolic depression, we hand-dissected ~10 tracheoblasts from individual progenitor clusters of both wandering L3 larvae and white pupae (0 h APF) (Supplementary Fig. 1a–c), performed RNA sequencing (RNA-seq), and compared transcriptomes of the preparations. Characterization of differentially expressed genes (DEGs) by a DAVID cellular component Gene Ontology (GO) analysis identified metabolic pathways as the most abundant functional class in the larval and pupal datasets (Fig. 2a). The Hippo pathway was also abundantly represented (Fig. 2b, c), suggesting that expression levels for components of the metabolic and Hippo pathways change upon activation. We also isolated and analyzed tracheoblasts from fasting L3 larvae, and we compared their transcriptomes to normals. Genes in the Hippo and insulin/PI3K signaling pathways are among those that showed the most significant changes in fasting animals (Fig. 2d–f). Insulin/PI3K signaling coordinates cellular metabolism with nutritional states[35], and the larval–pupal and normal-fasting datasets shared 704 DEGs in common, of which many are genes of the Hippo and metabolic pathways (Fig. 2g).

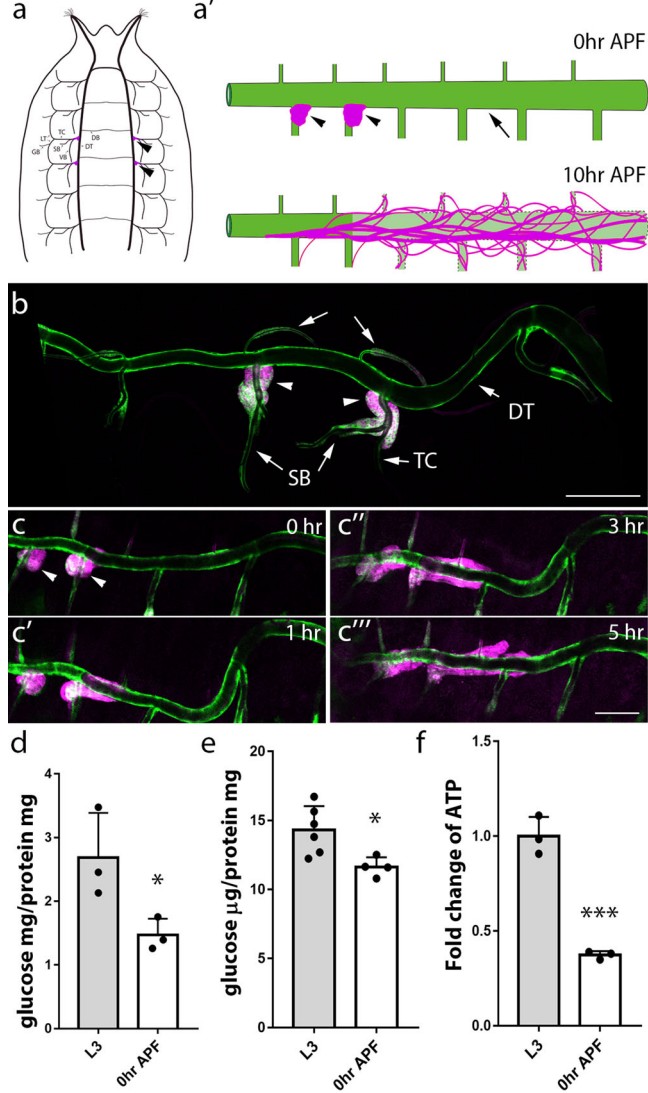

**Fig. 1 The activated tracheal progenitors migrating during pupariation.**
**a** Schematic diagram depicting the localization of progenitors (magenta) and trachea in a white pupa (0 h APF). Arrows denote dorsal trunk (DT), dorsal branch (DB), transverse connective (TC), visceral branch (VB), spiracular branch (SB), lateral trunk (LT), and ganglionic branches (GBs). Arrowheads point to tracheal progenitors (magenta). **a′** Schematic representation of trachea (green) and progenitors (magenta) at 0 h APF or 10 h APF. Activated progenitors move posteriorly along the dorsal trunk. The posterior tracheal branches outlined by dashed lines are replaced by tracheal progenitors. **b** The larval tracheal branches are visualized by membrane-tethered GFP under the control of *Dl-Gal4* driver. Arrows denote dorsal trunk (DT), dorsal branch (DB), transverse connective (TC) and spiracular branch (SB). Arrowheads point to tracheal progenitors (magenta) that reside on Tr4 and Tr5 metameres. **c–c′′′** Migration of tracheal progenitors (magenta, arrowheads) along the dorsal trunk at indicated ages. **b**, **c** Three independent experiments were repeated with similar results. **d** Bar graph represents the systemic level of glucose in L3 and 0 h APF flies ($n = 60$ for all groups). Three biological replicates were performed. *$p = 0.0478$. **e** Bar graph represents the level of glucose in the trachea of L3 larvae ($n = 120$) and white pupae (0 h APF) ($n = 80$). Four biological replicates were performed. *$p = 0.0205$. **f** Bar graph plots the relative abundance of tracheal ATP in L3 larvae and white pupae (0 h APF) ($n = 60$ for all groups). Three biological replicates were performed. **$p = 4.63e{-4}$. **d–f** Data are presented as mean values ± SD. Unpaired two-tailed *t*-test was used for all statistical analyses. No adjustments were made for multiple comparisons. Scale bars: 150 μm (**b–c′′′**). Genotype: (**b–c′′′**) *UAS-CD8:GFP; Dl-Gal4, P[B123]-RFP-moe*. Source data for (**d**, **e**, **f**) are provided as a Source Data file.

binding domain was either the forkhead-associated domain 1 (FHA1) or the phosphotyrosine-binding domain Src homology 2 (SH2) (Supplementary Fig. 2).

SPARK reporters for InR, Akt, AMPK were tested in 293 T cells. GFP droplets were observed under normal culture conditions, but were absent under similar conditions in which the substrate had alanine substituted for residues phosphorylated in InR (tyrosine), Akt (serine), and AMPK (threonine) (Supplementary Fig. 3a–f). This result is consistent with the idea that phosphorylation of these residues was responsible for kinase activity-dependent phase separation of the reporter. To validate the specificity and efficacy of the InR, Akt, and AMPK SPARK reporters, we next generated fly lines containing transgenes encoding InR-SPARK, Akt-SPARK and AMPK-SPARK, and expressed these transgenes in trachea under the control of *btl*-Gal4. GFP droplets were abundant in L3 trachea of the three reporter lines, but not in animals with reduced function of the respective kinases (Supplementary Fig. 3i–n). In pupariating larvae, the numbers and size of InR-SPARK and Akt-SPARK droplets were also reduced, while the number and size of AMPK-SPARK droplets increased (Fig. 2i, i′, j, j′, k, k′, q). These results are consistent with the idea that InR upregulates Akt and antagonizes AMPK. Similarly, we observed that in starved larvae, InR-SPARK and Akt-SPARK droplets were reduced and AMPK-SPARK droplets increased (Fig. 2i′′, j′′, k′′). These results suggest that insulin pathway is suppressed in the trachea during larval–pupal transition.

To analyze the activity of the Hippo pathway, we monitored the expression of ex-lacZ, a reporter for Yki signaling. We observed that ex-lacZ expression increased in tracheal progenitors during the larval–pupal transition and upon starvation (Fig. 2l, m′′, p). Compared with L3 larvae, the white pupae with elevated level of ex-lacZ decreased number and size of InR-SPARK droplets, which represents the reduction of InR activity (Fig. 2n–q). In sum, these results suggest that insulin and Hippo

To analyze the Insulin pathway further, we utilized the tGPH reporter in which the pleckstrin homology (PH) domain of the *Drosophila* general receptor for phosphoinositides-1 (GRP1) is fused to GFP. We monitored the membrane-associated fluorescence of tGPH, an indicator of PI3K activity[35]. Whereas membrane localization of tGPH was robust in larval tracheal cells (Fig. 2h), white pupae and starved larvae had less plasma membrane and enhanced nuclear fluorescence (Fig. 2h′, h′′). These results suggest that insulin pathway activity decreases in the trachea during pupariation.

Next, we probed for the activity of pivotal components the insulin pathway to understand the kinetics of the signaling cascade. In *Drosophila*, activation of the insulin receptor InR, a tyrosine kinase, stimulates kinases in the Akt and AMPK-FOXO pathways. We generated reporters that use the "separation of phases-based activity reporter of kinase" (SPARK) technique, which endows reporters with high sensitivity, large dynamic range and fast kinetics[36]. Phase separation and formation of droplets are achieved by the combination of multivalency and kinase activity-dependent protein–protein interaction (PPI). Multivalency is mediated by homo-oligomeric tags (HOTags). To generate kinase activity-dependent PPI, two protein chimeras were made: a consensus kinase activity-sensing motif fused to EGFP followed by Hotag3, and a phosphoserine/threonine-binding domain fused to Hotag6. The phosphoserine/threonine-

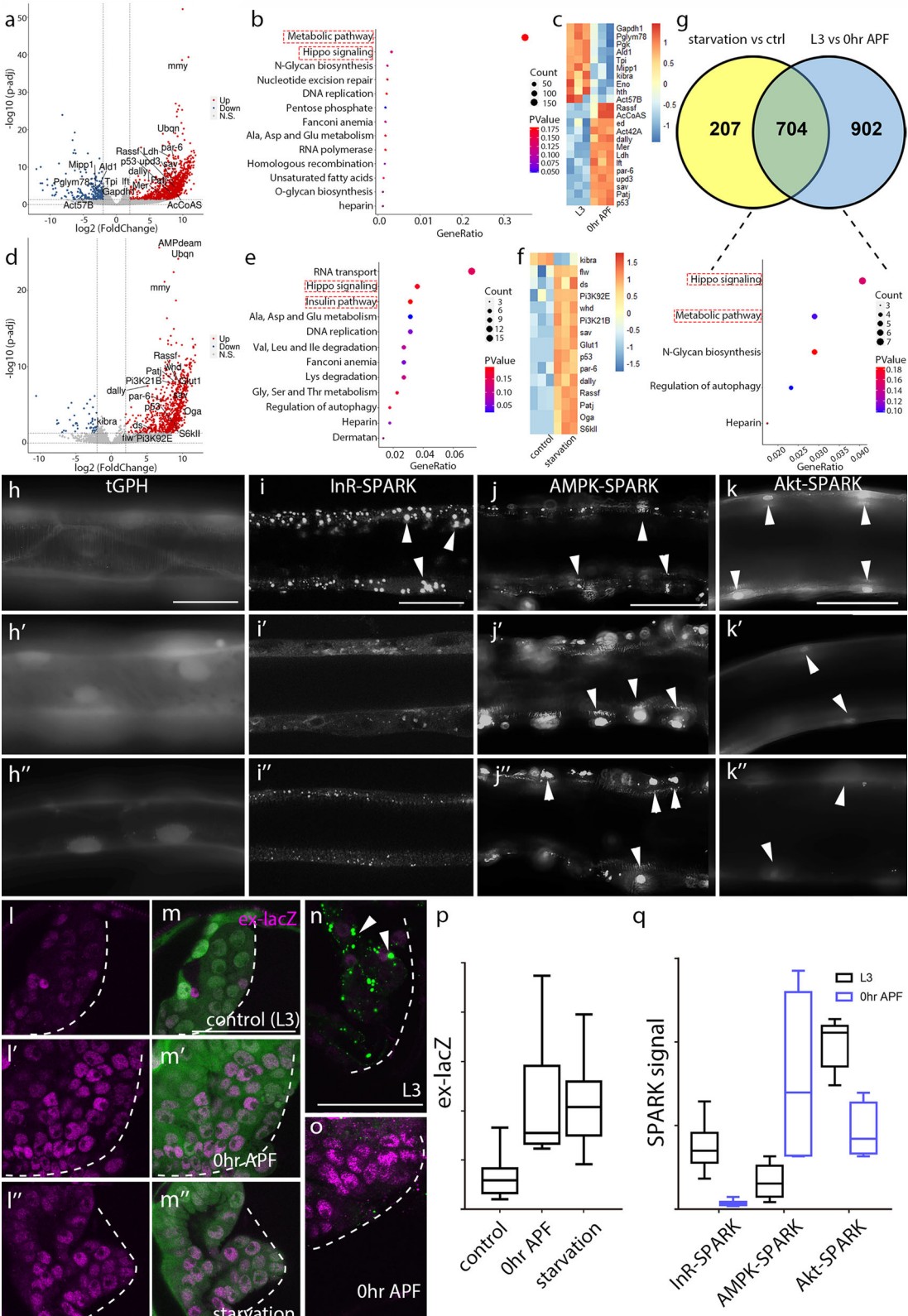

signaling respond to metabolic depression during larval–pupal transition when progenitors are activated, and that the activity of Yki inversely correlates with insulin signaling.

**Dependence of tracheal progenitors on Yki activity.** Cells of the L3 DT are post-mitotic and retain the potential to re-enter the cell cycle[37,38], but during puparium formation, tracheal progenitors divide. We identified cycling progenitors by EdU incorporation in both Tr4 and Tr5 at 0 h APF and observed that the number of EdU-labeled cells declined to low amounts by 150 min APF (Fig. 3a–d). Live imaging using the cell cycle indicator Fucci, which labels cells in the S/G2 and M phases with green fluorescence[39] confirmed the results of EdU labeling and showed

**Fig. 2 The activities of insulin pathway and Hippo pathway in larval–pupal transition and starvation conditions. a** Volcano plot of RNA-seq showing the comparison of gene expression profiles of tracheal progenitors in L3 larvae and white pupae during larval–pupal transition. **b** GO analysis reveals top functional clusters among the differentially expressed genes. **c** Heatmap showing the differential expression of genes in pupae relative to L3 larvae. **d** Volcano plot of RNA-seq showing differentially regulated genes in tracheal progenitors by starvation. **e** GO analysis reveals top functional clusters among the differentially expressed genes. **f** Heatmap showing the differential expression of genes by starvation. **g** Venn plot of differentially expressed genes (DEGs) in tracheal progenitors from starved L3 larvae and 0 h APF white pupae, compared to control L3 larvae. **h–m″** Dynamic activation of reporters in L3 larvae (**h, i, j, k, l, m**), 0 h APF pupae (**h′, i′, j′, k′, l′, m′**) and L3 larvae post-starvation (**h″, i″, j″, k″, l″, m″**). **h** Four independent experiments were repeated with similar results. **l–m″**, Staining tracheal progenitors of *btl > GFP* in L3 larvae (**l, m**), 0 h APF pupae (**l′, m′**) and L3 larvae post-starvation (**l″, m″**) with α-lacZ antibodies (magenta). **m–m″** Merge images. **n, o,** Merge images showing lnR-SPARK droplets (arrowheads, green) and ex-lacZ (magenta) in L3 larvae (n) and white pupae (o). Scale bars: 50 μm (**h–o**). **p** Box plot represents the relative level of ex-lacZ normalized to GFP signal from *btl > GFP*. 0 h APF ($n = 10$; $p = 1.38e\text{-}4$) and starvation ($n = 11$; $p = 9.07e\text{-}6$). $n = 15$ control flies. Four biologically independent experiments were performed. **q** Box plot showing the relative signals of SPARK reporters. InR-SPARK ($p = 2.02e\text{-}9$), AMPK-SPARK ($p = 9.38e\text{-}4$) and Akt-SPARK ($p = 1.61e\text{-}3$). $n = 16, 12, 10, 6, 5, 4$ from the left to the right. Three biologically independent experiments were performed. **p, q** Unpaired two-tailed *t*-test was used for all statistical analyses. Results are presented as median with minima and maxima. 25th–75th percentile (box) and 5th ~ 95th (whiskers) are indicated in the box plots. No adjustments were made for multiple comparisons in this figure. Source data for (**p, q**) are provided as a Source Data file.

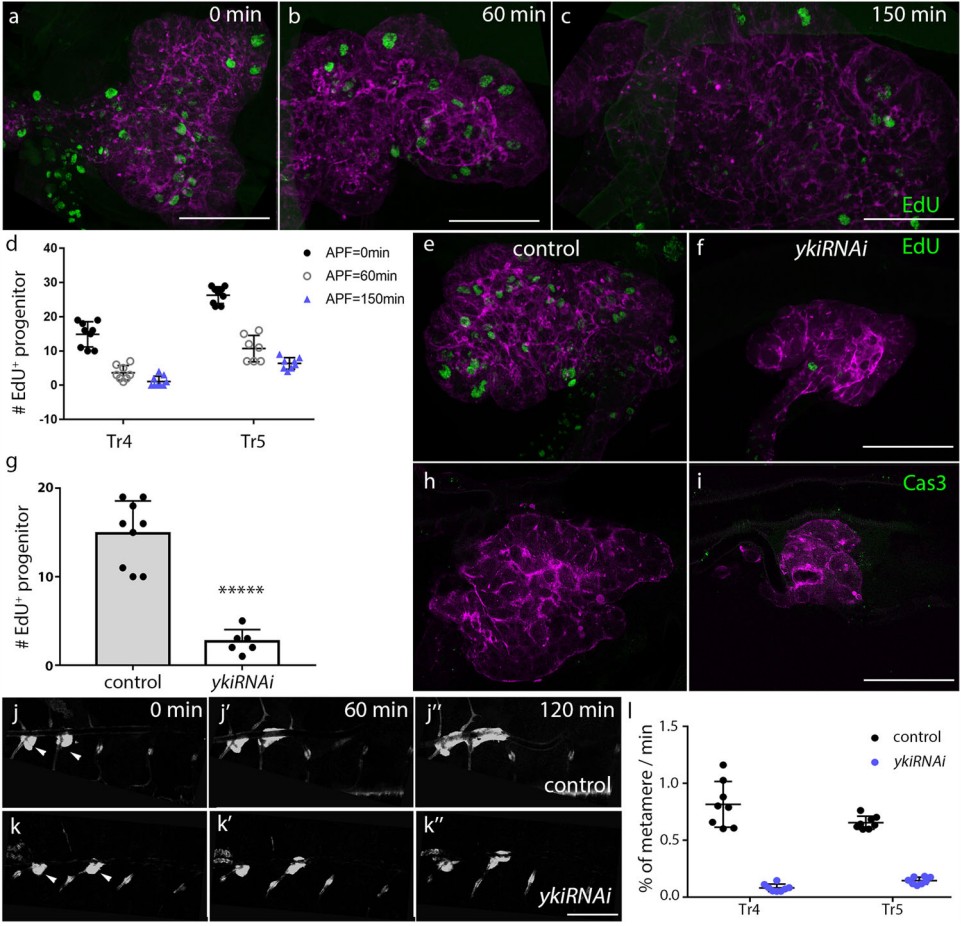

**Fig. 3 Yki signaling is required for the proliferation and migration of tracheal progenitors. a–c** The incorporation of EdU in the tracheal progenitors labeled by P[B123]-RFP-moe of pupae of indicated ages APF. **d** Scatter plot showing the number of EdU incorporation in Tr4 and Tr5 progenitors. $n = 9, 9, 8, 7, 9, 8$ from the left to the right. **e–i** Expression of *ykiRNAi* decreased the proliferation of progenitor, but did not cause apoptosis. Staining tracheal progenitors of white pupae with antibodies against EdU (**e, f**) or cleaved Caspase3 (**h, i**). **g** Bar plot depicts the incorporated EdU in the Tr4 tracheal progenitors of control ($n = 9$) and *ykiRNAi* flies ($n = 6$). *****$p = 3.44e\text{-}6$. **h, i** Three independent experiments were repeated with similar results. **j–k″** Dependence of progenitor migration on Yki. Arrowheads point to tracheal progenitors. **l** Scatter plot showing the velocity of migrating progenitors. $n = 8$ for each group. Error bars represent standard deviation. *p* value: Tr4 ($p = 7.42e\text{-}8$) and Tr5 ($p = 2.86e\text{-}12$). **d, g, l** Three biologically independent experiments were performed. Results are presented as mean values ± SD. No adjustments were made for multiple comparisons. Unpaired two-tailed *t*-test was used for all statistical analyses in this figure. Scale bars: 30 μm (**a–c, e, f, h, i**) and 300 μm (**j–k″**). Genotypes: (**a–c**) *P[B123]-RFP-moe/+*; (**e, h, j–j″**) *btl-Gal4/+*; *P[B123]-RFP-moe/tub-Gal80^{ts}*; (**f, i, k–k″**) *btl-Gal4/ UAS-ykiRNAi; P[B123]-RFP-moe/tub-Gal80^{ts}*. Source data for (**d, g, l**) are provided as a Source Data file.

that proliferative progenitors resided in the niche (Supplementary Movie 2).

To determine if Yki activity is involved in the behavior of tracheal progenitors, we expressed RNAi directed against *yki* in the trachea with *btl*-Gal4. Expression of ex-lacZ was reduced in these flies (Supplementary Fig. 4a, b'), the volume of the progenitor population was relatively small, and at 0 h APF, EdU incorporation was reduced in niche progenitors (Fig. 3e–g). Consistent with a non-apoptotic role of Yki in embryonic trachea[24], expression of *ykiRNAi* did not increase apoptosis of tracheal progenitors (Fig. 3h, i). Thus, the reduction of progenitors was caused by reduced proliferation. Upregulation of Yki activity either under conditions of RNAi expression targeted to Wts, a negative regulator of Yki, or expression of constitutive active form of Yki (YkiS168A), increased EdU incorporation in the tracheal progenitors and increased the number of progenitors (Supplementary Fig. 5a–c, f).

In order to evaluate the function of Yki in the motility of tracheal progenitors, we imaged tracheal progenitors in control and Yki-depleted conditions. Knockdown of *yki* reduced the velocity of migrating progenitors (~0.2x), and reduced the extent of progenitor migration (Fig. 3j–l, Supplementary Movie 3). In sum, these results are consistent with the idea that Yki signaling has integral roles in the proliferation and migration of tracheal progenitors.

**Insulin signaling negatively regulates Yki signaling**. We next investigated the role of insulin signaling. We compared the transcriptomes of tracheal progenitors from control preparations to preparations from animals that expressed a dominant negative mutant insulin receptor, InR[DN], under the control of *btl*-Gal4 (Supplementary Fig. 1d). Expression of *InR[DN]* reduced insulin/ PI3K activity (Supplementary Fig. 4c, d), and changed expression of Hippo pathway components (Fig. 4a–c). To verify these responses, we examined ex-lacZ reporter expression in these conditions of *InR[DN]* expression and observed that it increased in L3 tracheal progenitors (Fig. 4d, e'). Conversely, expression of ex-lacZ was reduced in larvae treated with exogenous insulin, consistent with the idea that Yki activity is inhibited by insulin signaling (Fig. 4f, g', j). To directly monitor the activity of Yki, we constructed a SPARK sensor for YAP/Yki (referred to here as YAP-SPARK) that has both a consensus HXRXXS motif that is conserved in fly Yorkie and human YAP[40] and a companion 14-3-3ζ fragment (Supplementary Fig. 2). Because phosphorylation of YAP inhibits its activity by retaining YAP in the cytoplasm[41], the presence of YAP-SPARK droplets indicates the phosphorylation status of YAP/Yki and inversely correlates with YAP/Yki activity. The specificity and efficacy of YAP-SPARK was established by showing that YAP-SPARK droplets diminished in cultured cells expressing a serine-to-alanine substitution and in flies expressing *wtsRNAi* (Supplementary Fig. 3g, h, o, p). Tracheal progenitors in YAP-SPARK-expressing flies treated with insulin had larger number and size of GFP droplets than controls, which is in accordance with reduction of ex-lacZ signal, suggesting that Yki activity is suppressed by the presence of insulin (Fig. 4h–k).

To assess the role of insulin signaling in Yki-dependent progenitor migration, progenitors were monitored in the presence and absence of exogenous insulin. We adapted an ex vivo culture condition that was developed for imaginal disc explants[42], and observed that tracheal progenitors moved along the DT (Fig. 5a–d, g). However, migration of progenitors was vastly impaired in the presence of exogenous insulin (Fig. 5e–g), and expression of *InR[DN]* triggered precocious migration of tracheal progenitors in white pupae (0 h APF) (Fig. 5h, i). These results collectively suggest that insulin might regulate Yki-mediated progenitor behavior by antagonizing Yki signaling.

**The phosphorylation of Yorkie by Akt**. To search for the kinase that phosphorylates Yki in the tracheoblasts, in vitro phosphorylation experiments were performed using recombinant GST-Yki and purified kinases. Since serine phosphorylation-mediated regulation of Yki and its interaction with protein 14-3-3 is well documented in multiple development contexts (Fig. 6a), we surveyed the kinases in the insulin pathway that target serine/ threonine residues, namely Akt, AMPK and PI3K. Phosphorylation of Yki was detected in the presence of Akt, but not with AMPK or PI3K (Fig. 6b). Yki and its homolog YAP contain the conserved HXRXXS motif (Fig. 6a), in which phosphorylation of Ser168 leads to cytoplasmic retention. Having determined that phosphorylation of Ser168 in the SPARK reporter was dependent on insulin signaling (Fig. 4h, i and Supplementary Fig. 2), and that the serine-to-alanine substitution in YkiS168A generated a constitutive active Yki phenotype (Supplementary Fig. 5c, f), we sought to determine whether Akt phosphorylates Yki at Ser168. Figure 6c shows that Akt was unable to phosphorylate the mutant YkiS168A, consistent with the idea that Akt phosphorylation occurs at Ser168 site.

We generated several observations consistent with the idea that phosphorylation of Yki is dependent on Akt. First, we expressed FLAG-tagged Yki in trachea under the control of *btl*-Gal4. Phosphorylation of FLAG-Yki was detected by an antibody that recognizes phospho-Akt substrates, but not by an antibody against phospho-AMPK substrates (Fig. 6d). Phosphorylation of Yki by Akt was eliminated in the mutant YkiS168A (Fig. 6e). Second, expression of *UAS-Akt* increased the size and number of YAP-SPARK droplets, suggesting that phosphorylation of Yki is elevated by the upregulation of Akt (Fig. 6f, g, j). Finally, in accordance with the inhibitory role of insulin in Yki signaling, overexpression of Akt in L3 animals reduced EdU incorporation (Fig. 6h, i, k) and progenitor migration (Fig. 6l–n). In sum, these observations indicate that Akt acts upstream of Yki in vivo and impedes Yki-dependent processes.

**Genes regulated by Yki in the trachea**. To further investigate the underlying molecular mechanism of Yki-regulated tracheal progenitor migration and proliferation, genomic chromatin immunoprecipitation (ChIP-seq) was performed to identify loci bound by Yki in trachea. To associate ChIP-seq peaks with putative gene targets, a peak-to-gene distance cutoff was required. Of total 10,512 peaks, 95% were located within 20 kb of transcription start sites (TSSs) (Supplementary Fig. 6). Analysis of the location of the peaks relative to the closest genes revealed that 92% of the peaks were enriched either in promoter regions or within gene bodies (Supplementary Fig. 6). 66% of peaks (6966 peaks) reside near the 5′ ends of annotated genes, namely in the promoter regions, first exon and first introns.

Consistent with previous ChIP-seq experiments for malpighian tubules and imaginal discs[26,43], the peaks we observed included multiple Yki binding sites in the promoter and intronic regions of *expanded* (*ex*) and *Diap1*, which are well-characterized direct target genes of Yki (Fig. 7a). Other Yki targets are regulators in cell migration, cell cycle, and cell adhesion (Fig. 7a, b), and include *hnt* (Fig. 7a), which is a regulator of cell cycle in follicle cells[44]. Yki occupancy was also pronounced in the promoter region of two heparan sulfate proteoglycans (HSPGs), Dally and Dally-like protein (Dlp) (Fig. 7a), which together with Ex and Diap1 belong to the cell migration GO cluster (Fig. 7b). Significant ChIP-seq peaks were observed in the region of *N-cadherin* (*Ncad*), a cell adhesion molecule that promotes cell migration (Fig. 7a). The GO cluster of cell adhesion also comprises a component of extracellular matrix, Matrix metalloproteinase-1 (MMP-1) (Supplementary Fig. 7k, k').

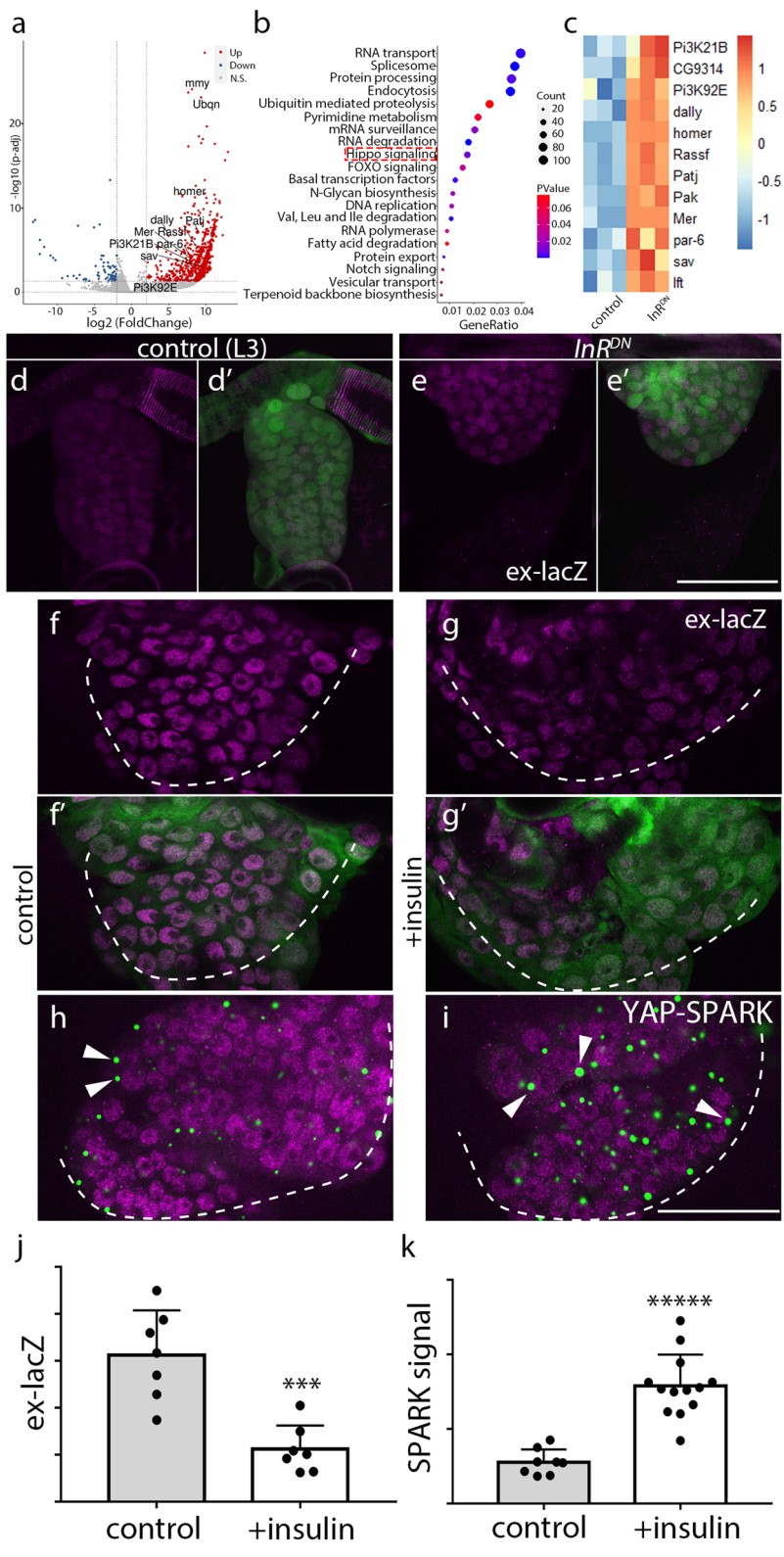

Relative enrichment was normalized with input (Supplementary Fig. 8a). ChIP-qPCR experiments further confirmed enrichment of Yki in promoter and/or regulatory regions of aforementioned putative targets (Supplementary Fig. 8b).

Analysis of motifs enriched in Yki-associated chromatin suggests that Yki-dependent transcription in the trachea involves recruitment of the transcription factor Trithorax-like (Trl), which is involved in cell division, Kruppel (Kr), which functions in tubule differentiation, Aef1, and Scalloped (Sd), which partners with Yki[45] (Fig. 7c). In addition, 60% of DEGs from RNA-seq of tracheal progenitors are downregulated by expression of *ykiRNAi* (Fig. 7d). More than 60% of DEGs that were identified by RNA-seq of trachea progenitors are also high confidence gene targets identified by the Yki ChIP-seq experiment (Fig. 7e).

**Fig. 4 Insulin signaling inhibits the activity of Yki signaling. a–c** Differential expression profiling in tracheal progenitors of $InR^{DN}$ pupae. **a** Volcano plot of RNA-seq showing differentially regulated genes up- and downregulated genes with four-fold and higher changes (upregulated genes in red; downregulated genes in blue) in $InR^{DN}$ compared with control. Complete differentially expressed genes (DEGs) from RNA-seq results were listed in Supplementary Fig. 1d. **b** GO analysis reveals top functional clusters among the differentially expressed genes. Hippo signaling pathway denoted in dashed box is in the list with high enrichment score. **c** Heatmap showing the differential expression of genes from two top enriched GO groups: Hippo pathway and FOXO pathway in $InR^{DN}$ relative to control. **d–e′** Expression of $InR^{DN}$ promoted the activity of Yki. Staining tracheal progenitors of $btl > GFP$ in control (**d, d′**) and in the presence of $InR^{DN}$ (**e, e′**) with α-lacZ antibodies (magenta). Three independent experiments were repeated with similar results. **d′, e′**, Merge images. **f–i** Elevation of insulin perturbs Yki signaling. **f–g′** The expression of ex-lacZ was decreased upon the administration of insulin. Merge images (**f′, g**). **h, i** The confocal images showing staining tracheal progenitors of ($btl > YAP\text{-}SPARK$) in control and insulin-treated pupae with α-lacZ antibodies. Merge images showing YAP-SPARK in green and ex-lacZ in magenta. The progenitors are outlined by dashed lines. Arrowheads denote the GFP droplets of YAP-SPARK sensors in tracheal progenitors. **j** Bar graph plots the expression of ex-lacZ normalized to GFP signal from $btl > GFP$. $n = 7$ for each group. ***$p = 3.63e\text{-}4$. **k** Bar graph represents SPARK signal in control ($n = 8$) and insulin-treated pupae ($n = 13$). *****$p = 3.13e\text{-}6$. **j, k** Three biologically independent experiments were performed. Results are presented as mean values ± SD. Unpaired two-tailed $t$-test was used for all statistical analyses. Scale bars: 50 μm (**d–i**). Genotypes: (**d, d′**) $ex\text{-}lacZ/+; btl\text{-}Gal4,UAS\text{-}GFP/ tub\text{-}Gal80^{ts}$; (**e, e′**) $ex\text{-}lacZ/UAS\text{-}InR^{DN}; btl\text{-}Gal4,UAS\text{-}GFP/tub\text{-}Gal80^{ts}$; (**f, f′, g, g′**) $ex\text{-}lacZ/+; btl\text{-}Gal4,UAS\text{-}GFP/ +$; (**h, i**) $btl\text{-}Gal4/ex\text{-}lacZ; UAS\text{-}YAP\text{-}SPARK+$. No adjustments were made for multiple comparisons in this figure. Source data for (**j, k**) are provided as a Source Data file.

To test whether candidate genes identified by ChIP-seq analysis depend on Yki, we monitored the candidate proteins using antibody and fluorescence tags. Expression of *ykiRNAi* decreased the apparent abundance of Hnt, Knirps (Kni) (Fig. 7f, g and Supplementary Fig. 7a, b') and Dally and Dlp (Fig. 7h–k') in tracheal progenitors. The effects on Dally and Dlp are consistent with the proposal that the Hippo pathway and these HSPGs are interdependent in wing discs[46]. We also examined other candidates in the cell migration GO term cluster. Rho1, a target of Hippo pathway in tumorigenesis[47], was significantly decreased by expression of ykiRNAi (Supplementary Fig. 7c, d'). Roundabout 2 (Robo2), a member of the Robo receptor family that mediates cell-cell interactions was reduced by knockdown of *yki* (Supplementary Fig. 7e, f'). Additionally, two matrix proteins, Serpentine (Serp) and Vermiform (Verm) were reduced by expression of *ykiRNAi* (Supplementary Fig. 7g–j'). The results of these experiments suggest that Yki promotes cell migration by regulating a variety of constituents of the extracellular matrix. Moreover, levels of Ncad, a cell adhesion molecule implicated by the ChIP-seq analysis, elevated during pupariation, which correlated with the motility of tracheal progenitors (Supplementary Fig. 9). the abundance of Ncad was severely lowered upon the reduction of Yki (Fig. 7l, m'). Matrix metalloproteinase 1 (Mmp1), another gene present in the cell adhesion GO term cluster, was downregulated by expression of *ykiRNAi* (Supplementary Fig. 7k, l'). Together, these results suggest that Yki activates various targets that function in cell cycle, cell migration and cell adhesion.

**Roles of Yki targets in cell cycle and migration**. We evaluated the roles of candidate Yki targets by RNAi-mediated downregulation. Expression of *hntRNAi* in the trachea decreased cell proliferation, as assessed by EdU incorporation (Fig. 8a–c), suggesting that Hnt is an essential factor in Yki-dependent proliferation. However, Dally, Serp, and Ncad that have roles in cell migration or cell adhesion did not affect the proliferation of tracheal progenitors, which suggests that Yki signaling controls progenitor proliferation and migration though distinct effectors (Fig. 8c). Expression of RNAi constructs directed against Dally, Dlp, and Ncad reduced tracheal progenitor migration, suggesting their important roles in the migration of tracheal progenitor (Fig. 8d–g", 8j, Supplementary Movie 3). The roles of the Yki binding partners Trl and Sd were also analyzed. Expression of *TrlRNAi* reduced migration and expression of *sdRNAi* blocked migration completely (Fig. 8h–j; Supplementary Movie 3). Proliferation of tracheal progenitors was also severely reduced by expression of *TrlRNAi* or *sdRNAi* (Supplementary Fig. 5d–f). Overall, these results suggest that distinct downstream targets are required for Yki-dependent proliferation and migration of tracheal progenitors.

## Discussion

Tissue growth and stem cell proliferation are tied to energy production and consumption (Lin et al., 2007), but there are contexts such as insect metamorphosis in which cell division and growth coincides with reduced metabolism. The fact that adult stem cells in holometabolous insects initiate programs of cell division and tissue remodeling when the animals stop feeding at metamorphosis appears to reverse the normal relationship between energy production and growth. This raises the possibility that signals generated by starvation that would otherwise arrest growth and development instead activate stem cells that are poised for growth, migration, and remodeling. We investigated the basis for adult stem cell activation at metamorphosis by studying *Drosophila* tracheal progenitors that have been shown to reconstitute and regenerate the tracheal system (Chen and Krasnow, 2014). Our findings show that insulin signaling senses the metabolic depression, reduces Akt-mediated phosphorylation of Yki, and activates Yki-dependent proliferation and migration of tracheal progenitor cells (Supplementary Fig. 10). It appears that the Yki signaling in the progenitor interferes with FGF pathway based on the following observations. First, the expression of *bnl* coincides with the activity of Yki (Supplementary Fig. 11a–e). Importantly, Bnl expression was reduced upon expression of *ykiRNAi* and was elevated upon upregulation of Yki activity by expressing constitutive active form of Yki (YkiS168A) or by overexpression of Yki (Supplementary Fig. 11f–j). Second, considerable peaks were detected in the region of *bnl* in ChIP-seq experiments with Yki antibodies, indicating the association of Yki (Supplementary Fig. 11k). Phenotypically, perturbation of FGF pathway by expression of dominant negative form of FGF receptor, $Btl^{DN}$, or by expressing RNAi against *branchless* (*bnl*) that encodes an FGF ligand phenocopied migration deficit caused by Yki abrogation, and such a defect can be partially rescued by induced expression of Yki (Supplementary Fig. 11l–n).

Hippo-Yki/YAP signaling influences stem cell proliferation and exerts growth control in developmental contexts and oncogenic condition[48–51]. In the canonical Hippo kinase cascade, the Hippo-Salvador (Sav) complex activates the Wts by phosphorylation. Extensive studies in *Drosophila* using genetic screen discovered a class of Hippo targets as well as regulators of Yki, such as Expanded (Ex), Kibra, Crumbs (Crb), Merlin (Mer) and RASSF[52]. The roles of Yorkie in mediating stem cell proliferation in concert with multiple pathways are identified by the investigation of *Drosophila* adult midgut whose homeostasis is

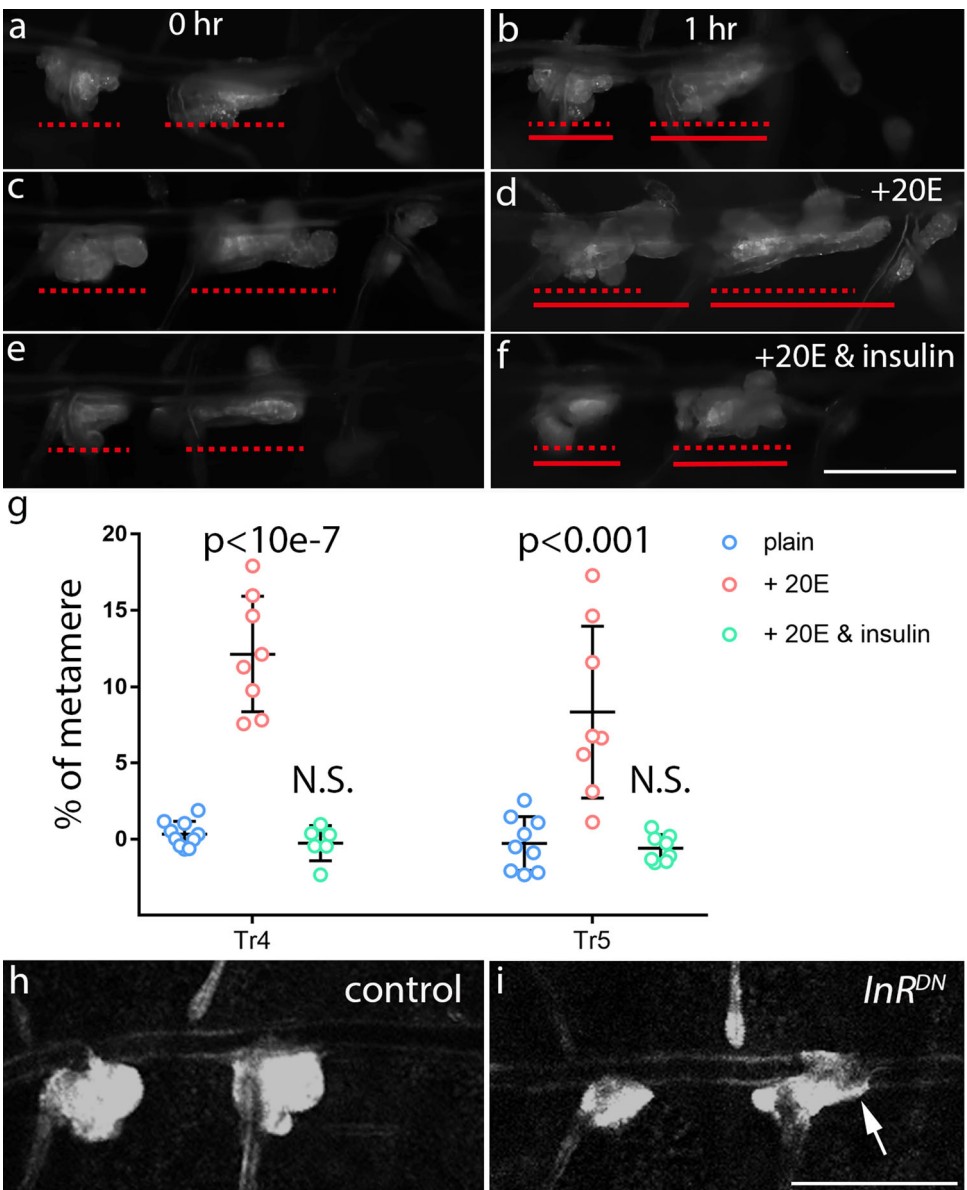

**Fig. 5 The effects of insulin on progenitor migration. a–d** An ex vivo culture condition with a key supplement of steroid hormone 20-hydroxyecdysone (20E) was sufficient to preserve the migration of tracheal progenitors. **e, f** Exogenous insulin prevents the migration of tracheal progenitors. Pupal trachea in Grace's Insect Medium (GIM) (**a, b**), ecdysone (20E) plus GIM (**c, d**) and ecdysone (20E) plus GIM with insulin (**e, f**). The positions of progenitors at 0 h APF and 1 h APF are indicated by dashed lines or solid lines, respectively. **g** Scatter plot depicting migration distance of tracheal progenitors. Five biologically independent experiments were performed. Tr4: plain (n = 10), 20E (n = 8; p = 4.47e-8), 20E&insulin (n = 6; p = 0.253). Tr5: plain (n = 9), 20E (n = 8; p = 5.48e-4), 20E&insulin (n = 6; p = 0.669). Data are presented as mean values ± SD. N.S. not significant. Unpaired two-tailed t-test was used for all statistical analyses. No adjustments were made for multiple comparisons. **h, i** Reduced insulin activity promotes migration of tracheal progenitors. Lowering insulin activity by expressing $InR^{DN}$ for 20 h in L3 triggered early migration of tracheal progenitors in white pupae. Five independent experiments were repeated with similar results. Arrow denotes the progression of progenitors on dorsal trunk. Scale bars: 200 μm (**a–f, h, i**). Genotypes: (**a–f**) P[B123]-RFP-moe/+; (**h**) btl-Gal4/+; tub-Gal80ts/ P[B123]-RFP-moe; (**i**) btl-Gal4/UAS-InRDN; tub-Gal80ts/ P[B123]-RFP-moe. Source data for (**g**) are provided as a Source Data file.

dependent upon the resident stem cells[50,53]. Similar to *Drosophila* intestine, the larval tracheae are also monolayer epithelial tubes, although less stratified and less in abundance. Unlike intestinal stem cells that are sparsely distributed along the basal side of the epithelium, tracheal progenitors reside in restricted anatomical locations before moving out of the niche. The proximity to cuticle and characteristic molecular signature that they possess permit the ease of histological analysis and imaging them in living animals. We particularly focus on the stem cell migration which is not applicable to midgut system and is not explored in other systems. By examining the signaling activity in the resident and migratory tracheal progenitors, we found that progenitors defective for Yki or its targets can not move and are unable to proliferate. Its activation depends on the decline of insulin pathway, which is in accordance with the notion that Yki receives input from multiple kinase pathways other than canonical Hippo signaling[22] and suggests that insulin is a signal that stimulates progenitor cells under metabolic depression. The results reported here pinpoint the integral roles of Yki in controlling the progenitor cell proliferation, which is consistent with precedents in

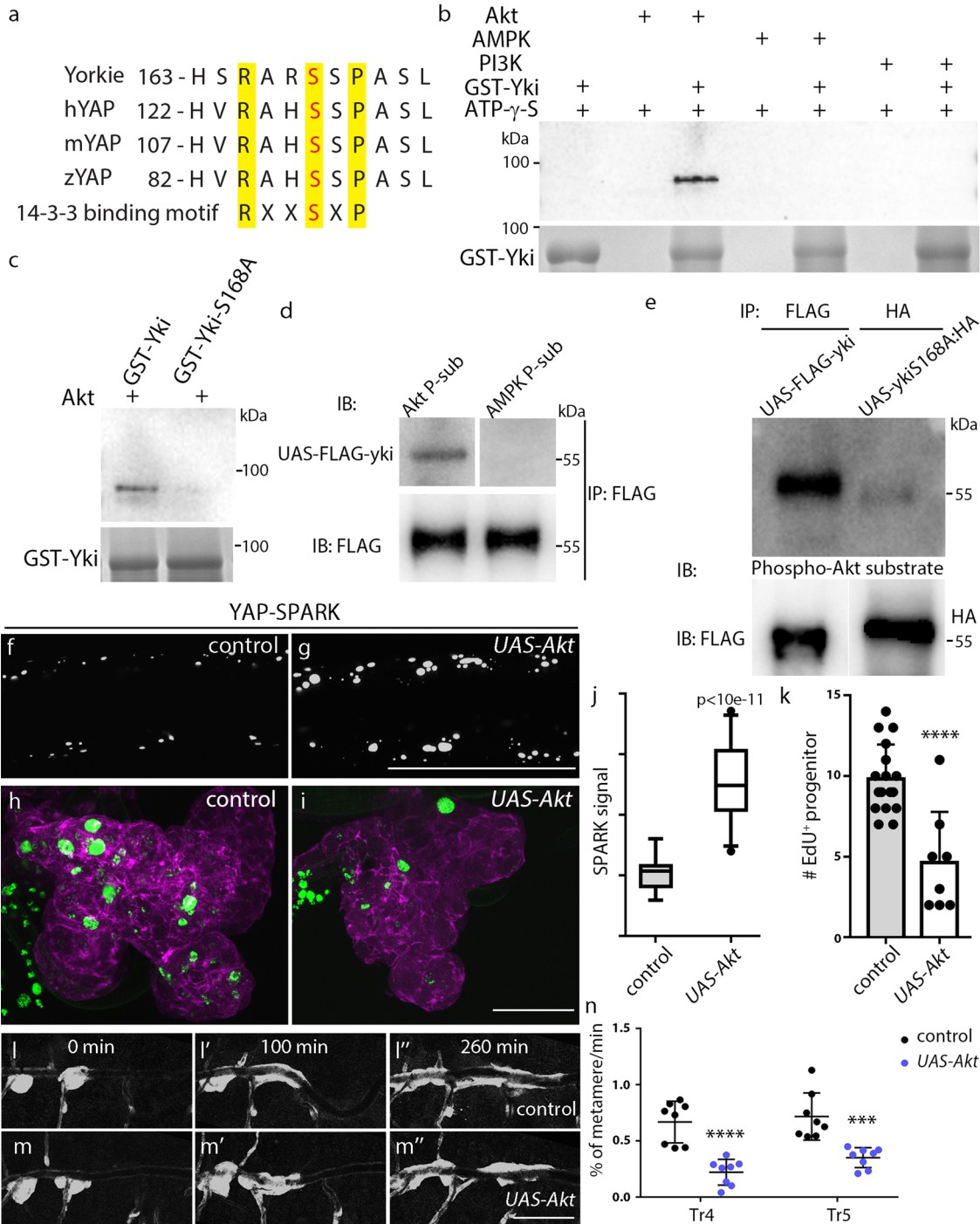

intestinal stem cells[53]. Its function in migration of progenitors suggests a link between mitotic division and deployment of stem cells. Furthermore, both processes are influenced by metabolic status. Among numerous targets of Yki, it affects cell division partly through regulating the expression of Hnt, while it promotes cell mobility by regulating components of extracellular matrix (ECM). Our results show that HSPGs, principal components of the ECM, MMP1 that exerts essential functions in remodeling of the ECM, and deacetylases of chitin, a polysaccharide in the ECM are controlled by Yki activity (Fig. 7h–k' and Supplementary Fig. 7g–l'), which might indicate that tracheal progenitor cells undergo the remodeling of the ECM and alter the composition of their ECM to initiate their migration and this process is dependent on Yki signaling.

Unlike Yki, its DNA-binding partner, Sd, is dispensable for normal growth in most *Drosophila* tissues. Here, we show that Sd depletion is reminiscent of *ykiRNAi* phenotype in several aspects, suggesting the participation of Sd in Yki-dependent transcription in the trachea. Other frequent partners of Yki in the trachea implicated by our ChIP-seq experiments include GAGA factor, Trl that links chromatin modification and transcriptional activation and functions in cell division, which is consistent with previous report that Yki is in complex with Trl and Brahma complex subunit Moira[26].

Studies in mammalian Hippo pathway show that multiple residues in YAP are accessible for various kinases, where Ser127 of YAP, the mammalian equivalent site of Yorkie Ser168, is critical for signal inactivation. Phosphorylation on it exposes the

**Fig. 6 Akt phosphorylates Yki in vitro and in vivo. a** Comparative analysis of *Drosophila* Yorkie, human (h), mouse (m) and zebrafish (z) YAP proteins. **b** The GST-Yki proteins are phosphorylated by Akt. The amount of each GST–YAP protein was detected by Coomassie staining. **c** Akt phosphorylates GST-Yki but not GST-YkiS168A. **d** Akt phosphorylates Yki in vivo. Immunoprecipitation assay of FLAG-Yki expressed in fly trachea. Phosphorylated proteins were detected using antibodies that recognize phospho-Akt substrates or phospho-AMPK substrates. **e** Yki is phosphorylated at Ser168 by Akt in the trachea. Western blot analysis of precipitated FLAG-Yki and YkiS168A-HA with antibodies against phospho-Akt substrates. **b–e** Three independent experiments were repeated with similar results. **f, g** Akt enhances signal of YAP-SPARK in the trachea. **h–m** Akt attenuates Yki-dependent activation of tracheal progenitors. **h, i** The incorporation of EdU in the tracheal progenitors of control (**h**) and *UAS-Akt* (**i**). **j** Box plot depicts the signal of YAP-SPARK reporter in control (*n* = 16) and *UAS-Akt*-expressed trachea (*n* = 31). Six biologically independent experiments were performed. Data are presented as median with minima and maxima. 25th–75th percentile (box) and 5th–95th percentile (whiskers) as well as outliers are indicated in the box plots. $p = 9.46e-12$. **k** Bar graph showing the number of EdU incorporation in control (*n* = 17) and *UAS-Akt*-expressed trachea (*n* = 8). Three biologically independent experiments were performed. Data are presented as mean values ± SD. ****$p = 6.38e-5$. **l–m″** Expression of *UAS-Akt* suppresses the migration of tracheal progenitors. **n** Scatter plot showing the velocity of migrating progenitors in control (*n* = 8; $p = 4.69e-5$) and *UAS-Akt*-expressed trachea (*n* = 8; $p = 4.53e-4$). Eight biologically independent experiments were performed. Data are presented as mean values ± SD. **j, k, n** Unpaired two-tailed *t*-test was used for all statistical analyses. No adjustments were made for multiple comparisons. Scale bars: 100 μm (**f, g**), 30 μm (**h, i**) and 300 μm (**l–m″**). Genotypes: (**d**) *btl-Gal4/+; UAS-FLAG-yki/+;* (**f**) *btl-Gal4/+; UAS-YAP-SPARK/tub-Gal80^ts^;* (**g**) *btl-Gal4/UAS-Akt; UAS-YAP-SPARK/tub-Gal80^ts^;* (**h, l–l″**) *btl-Gal4/+; P[B123]-RFP-moe/ tub-Gal80^ts^;* (**i, m–m″**) *btl-Gal4/UAS-Akt; P[B123]-RFP-moe/tub-Gal80^ts^.* Source data for (**j, k, n**) are provided as a Source Data file.

---

docking site for its binding partner, 14-3-3 proteins and results in cytoplasmic retention. The kinases recognize HXRXXS phosphorylation consensus motif[40]. The results in present study show that Akt targets this motif and phosphorylates Yki/YAP, suggesting a conserved interplay between insulin pathway and Yki signaling. The Akt-mediated phosphorylation of Yki might be one of the molecular events that connect energy sensing to the regulation of Yki signaling. The Yki-dependent activation of progenitor cells is controlled by insulin-driven Akt signaling, indicating that Akt has a role in restricting proliferation and migration of progenitor cells, although it is originally identified as an oncogene and promotes tumor progression.

Phosphorylation by distinct kinases are critical steps in cellular metabolism. Glucose modulates the activity of several proteins kinases including these described in present study[54,55]. Metabolic perturbation evokes ramified signaling networks that involve multiple kinase pathways. Spying on the activity of kinases remains challenging especially in live tissue, partially due to their fast dynamics and ephemeral feature[56–58]. Instead of analyzing fixed specimens with immunohistochemistry that provides steady-state abundance, but is incapable of presenting their kinetics with sensitivity and precision, we delineated the dynamics of kinase activity by direct imaging of physiological contexts that express genetically encoded reporters. These reporters featured as large dynamic range, ultrasensitivity, fast kinetics and reversibility in live animals unveil the decline of InR and Akt activity and upregulation of AMPK and Yki activity upon metabolic depression, which confirms a complex operational relationship between metabolic pathways and detangles their antagonism in real time (Fig. 2).

Starvation is generally considered to restrict tissue growth by halting the energy supply. However, due to their susceptibility to insulin and Yki signaling, progenitor cells are activated and accelerate their proliferation under metabolic deficit.

## Methods

**Drosophila husbandry.** Flies were reared on standard cornmeal and agar medium at 25 °C, unless otherwise mentioned. To generate SPARK transgenic flies, the fragment that comprised Hotag3-tethered substrate, EGFP and Hotag6-tethered companion segment was inserted into an attB site-containing pUAST vector. The construct was verified by DNA sequencing and was injected into y[1] M{vas-int.Dm}ZH-2A w[*]; P{CaryP}attP2 recipient flies following standard *Drosophila* transformation injection procedures by Core Facility of Drosophila Resource and Technology, SIBCB, CAS. See Key Resources Table for detailed information of flies used in this study. We obtained the following stains from Bloomington Drosophila Stock Center: *UAS-ykiRNAi* (34067), *UAS-yki.S168A* (28818), *UAS-InR^DN^* (8252), *UAS-Akt* (8191), *UAS-dlpRNAi* (34091), *UAS-bnlRNAi* (34572), *UAS-AMPK^DN^* (32112) and *UAS-ykiS168A:HA-GFP* (28816). The following flies were gained from

Vienna Drosophila RNAi Center: *UAS-ykiRNAi* (104523, 40497), *UAS-wtsRNAi* (106174), *yki-V5-Flag* (318237) and *dally:YFP* (115511). The following strains were obtained from Tsinghua Stock Center: *UAS-ykiRNAi* (THU0579), *UAS-sdRNAi* (THU2534), *UAS-AktRNAi* (THU0552), *UAS-NcadRNAi* (THU2665) and *UAS-TrlRNAi* (THU03912.N). *Dl-Gal4* was obtained from Edan Foley, *ex-lacZ* was from Jin Jiang, *UAS-hntRNAi* was from Howard Lipshitz, *UAS-dallyRNAi* was from Hiroshi Nakato, and *UAS-Flag-yki* was from Richard Fehon.

**Live imaging of *Drosophila* trachea.** White pupae of *Drosophila* (0 h APF) were briefly washed and cleaned. Pupae were mounted in halocarbon oil 700 (Sigma). Next, pupae were mixed well with oil, positioned with forceps and rolled so that a single dorsal trunk of the trachea is up for optimal imaging of Tr4 and Tr5 metameres. Then, pupae were immobilized by a 22 × 30 mm No.1.5 high precision coverslip spaced by vacuum grease. The time-lapse images were captured by an LSM Zeiss 900 inverted confocal laser scanner microscope with 405 nm, 488 nm, 561 nm and 640 nm wavelength lasers.

**Glucose assay.** Trachea dissected from wandering L3 larvae or white pupae 0 h APF ($N = 20$) were homogenized in 100 μL lysis buffer (APLLYGEN, #E1011) with protease inhibitor (Roche) on ice. After 10 min centrifugation (12,000 g, 4 °C), the supernatant was transferred into a new 1.5 mL EP tube. 20 μL supernatant will be used to measure protein contents (Bradford assay). 180 μL reaction buffer (APLLYGEN, #E1011) was added to supernatant and was then incubated at 37 °C for 30 min. The absorbance at 555 nm was measured by a Multiskan Sky Microplate Spectrophotometer (Thermo, #15748147). A glucose standard curve was used to calculate glucose content. The final glucose level was calculated relative to protein concentration.

**ATP assay.** Trachea were dissected from wandering L3 larvae or white pupae 0 h APF ($N = 20$) and homogenized in 100 μL PBS containing 4 mM EDTA and protease inhibitor (Roche) on ice. The sample was centrifuged at 12,000 g for 10 min at 4 °C. The supernatant was transferred into a new tube and boiled for 5 min. 20 μL supernatant was diluted in 80 μL double distilled water and then mixed well with 100 μL CellTiter-Glo® Reagent (Promega, #G7573) in a 96-well plate. The plate was incubated at room temperature for 10 min and was then measured with a Varioskan Flash (Thermo, #5250040). Each reading was normalized to protein concentration.

**RNA sequencing of tracheal progenitors.** The L3 larvae or white pupae (0 h APF) were dissected in cold PBS and a single cluster of progenitors from Tr5 metamere were subjected to RNA extraction using the RNeasy Micro Kit from Qiagen (#74004). Total RNA from each sample was used for sequencing library preparation. Three biological replicates were performed for each genotype or treatment. The SMART-Seq v4 Ultra low input RNA Kit (Takara Bio) was used for first-strand and second strand cDNA synthesis and double-stranded cDNA end repair. Double strand cDNAs were purified using the AMPure XP from Beckman Coulter (#A63881), subjected to tagmentation and ligated to adaptors. Finally, the libraries were generated by PCR enrichment of the adaptor-ligated DNA. The concentration and quality of the constructed sequencing libraries were measured by using the Agilent High Sensitivity DNA Kit and a Bioanalyzer 2100 from Agilent Technologies. The libraries were submitted to 150 bp paired-end high throughput sequencing using Hiseq4000.

RNA-seq data analysis was performed using a super computer system equipped with multiple processors. The clean reads were mapped to the *Drosophila* genome sequence using Hisat2 with default parameters. The number of mapped reads were counted by featureCounts. Differential gene expression analysis was performed

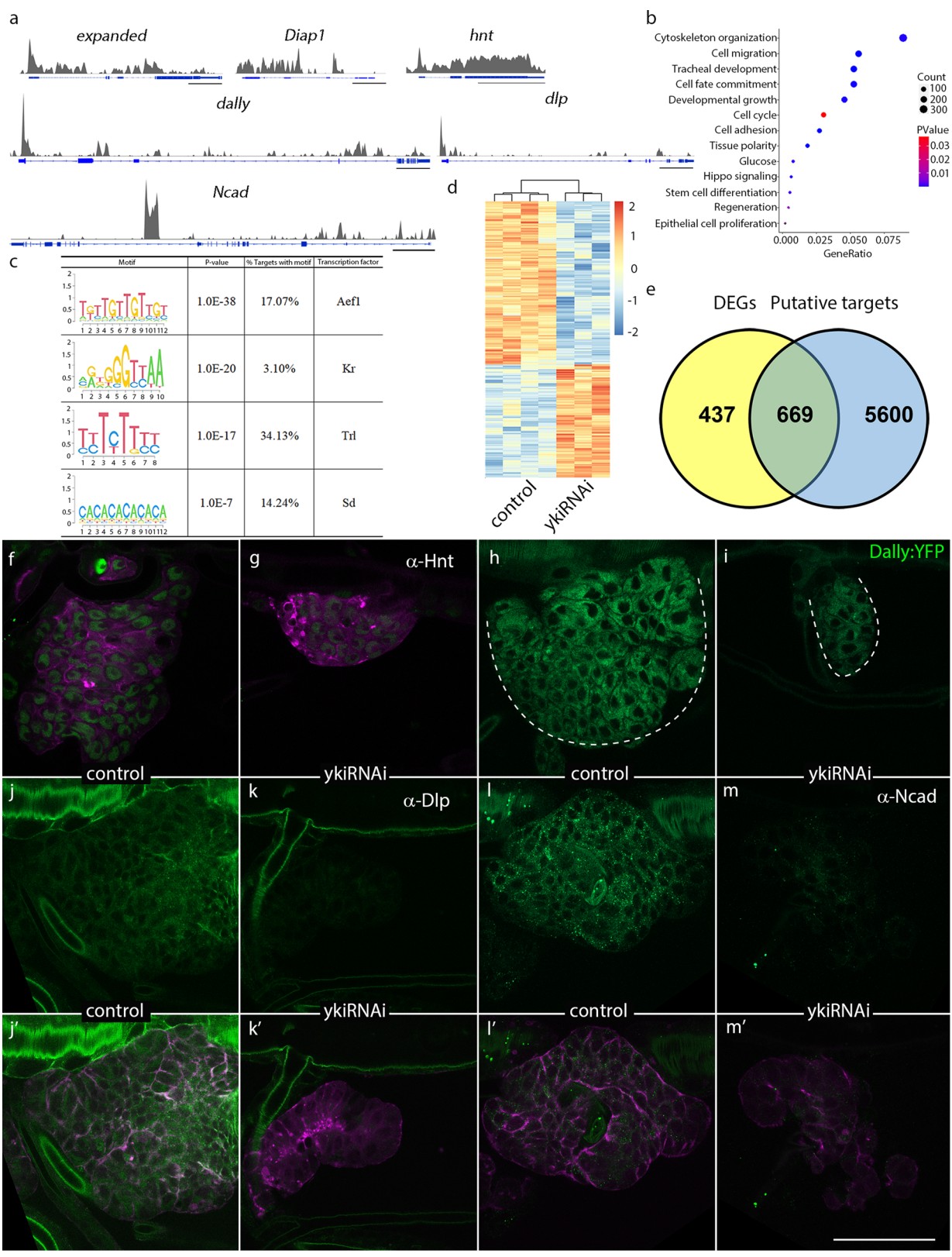

using the DESeq2 package. Adjusted *p* value <0.05 was used as the threshold to identify the differentially expressed genes. Gene ontology and KEGG pathway enrichment analyses for the differentially expressed genes were conducted using the Database for Annotation, Visualization and Integrated Discovery (DAVID).

**Starvation assay**. Flies were reared on standard cornmeal and agar medium at 25 °C till early L3 stage. The L3 larvae were transferred to a vial without medium at

25 °C for 5-hour starvation. The larvae were starved on filter paper (Ø10mm) soaked with distilled water.

***Gal80^{ts} inactivation***. The expression of btl-Gal4 was restricted by temperature sensitive *tub-Gal80^{ts}*. Larvae expressing *UAS-InR^{DN}*, *UAS-Akt*, *UAS-AktRNAi*, *UAS-ykiRNAi* or *UAS-HntRNAi* were raised at 18 °C and then were shifted to non-

**Fig. 7 Identification of gene targets of Yki in _Drosophila_ tracheal progenitors. a–c** Localization of Yki on chromosomes in trachea. **a** ChIP-seq peaks at loci regulated by Yki. Scale bar: 5 kb, except for _Ncad_ (10 kb). Enrichment was normalized input. **b** Bubble plot represents GO analysis showing top functional clusters among gene targets. **c** Cis-regulatory elements and corresponding transcription factors in the gene targets of Yki. **b, c** No adjustments were made for multiple comparisons. **d** Heatmap depicting differentially expressed genes (DEGs) in tracheal progenitors between control and _ykiRNAi_ flies. **e** Venn plot showing comparison between DEGs from RNA-seq and gene targets of ChIP-seq with Yki antibody. **f–m′** Validation of gene targets of Yki ChIP-seq. Two independent experiments were repeated with similar results. **f, g** Staining tracheal progenitors with α-Hnt antibody. **h, i** GFP fluorescence of Dally:YFP in the tracheal progenitors of control and _ykiRNAi_ flies. **j–m′** Staining tracheal progenitors of control (**j, j′, l, l′**) and _ykiRNAi_-expressing flies (**k, k′, m, m′**) with α-Dlp and α-Ncad antibodies. **j′, k′, l′, m′,** Merge images. Genotypes: (**f, j, j′, l, l′**) _btl-Gal4/+; P[B123]-RFP-moe/tub-Gal80ts_; (**g, k, k′, m, m′**) _btl-Gal4/UAS-ykiRNAi; P[B123]-RFP-moe/tub-Gal80ts_; (**h**) _btl-Gal4/+; dally:YFP/tub-Gal80ts_; (**i**) _btl-Gal4/ UAS-ykiRNAi; dally:YFP/tub-Gal80ts_. Scale bars: 50 μm (**f–m′**).

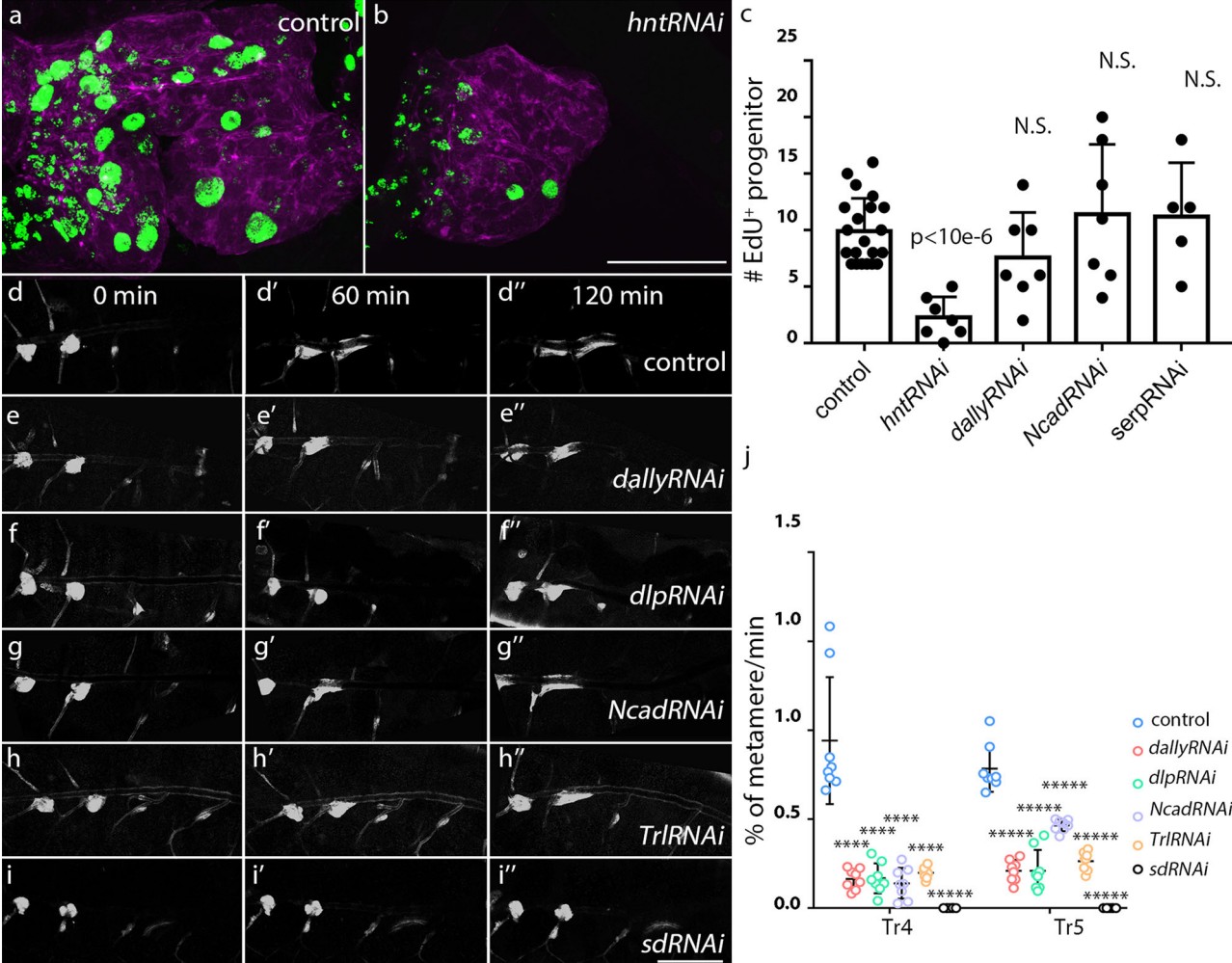

**Fig. 8 The targets of Yki regulate the proliferation and migration of tracheal progenitors. a–c** The proliferation of tracheal progenitors requires Hnt. The incorporation of EdU in the tracheal progenitors of control (**a**) and _hntRNAi_ (**b**). **c** Bar graph plots the number of EdU incorporation in control ($n = 21$), _hntRNAi_ ($n = 7$; $p = 7.39e-7$), _dallyRNAi_ ($n = 7$; $p = 0.106$), _NcadRNAi_ ($n = 7$; $p = 0.380$), _serpRNAi_ ($n = 5$; $p = 0.437$). Three biologically independent experiments were performed. N.S. not significant. **d–i″** Dependence of tracheal progenitor migration on Dally (**d–d″**), Dlp (**e–e″**), Ncad (**g–g″**), Trl (**h–h″**) and Sd (**i–i″**). Confocal images showing the localization of tracheal progenitors at 0 h APF (**d, e, f, g, h, i**), 1 h APF (**d′, e′, f′, g′, h′, i′**) and 2 h APF (**d″, e″, f″, g″, h″, i″**). **j** Scatter plot showing the velocity of migrating progenitors. Eight biologically independent experiments were performed. $n = 8$ for each genotype. Tr4: _dallyRNAi_ ($p = 2.77e-5$), _dlpRNAi_ ($p = 3.38e-5$), _NcadRNAi_ ($p = 2.38e-5$), _TrlRNAi_ ($p = 4.09e-5$) and _sdRNAi_ ($p = 3.02e-6$); Tr5: _dallyRNAi_ ($p = 2.05e-8$), _dlpRNAi_ ($p = 2.39e-7$), _NcadRNAi_ ($p = 9.76e-6$), _TrlRNAi_ ($p = 5.32e-8$) and _sdRNAi_ ($p = 9.47e-11$). **c, j** Results are presented as mean values ± SD. Unpaired two-tailed _t_-test was used for all statistical analyses. No adjustments were made for multiple comparisons. Scale bars: 30 μm (**a, b**) and 300 μm (**d–i″**). Genotypes: (**a**) _btl-Gal4/+; P[B123]-RFP-moe/tub-Gal80ts_; (**b**) _btl-Gal4/UAS-hntRNAi; P[B123]-RFP-moe/tub-Gal80ts_; (**d–d″**) _btl-Gal4/+; P[B123]-RFP-moe/+_; (**e–e″**) _btl-Gal4/+; P[B123]-RFP-moe/UAS-dallyRNAi_; (**f–f″**) _btl-Gal4/+; P[B123]-RFP-moe/UAS-dlpRNAi_; (**g–g″**) _btl-Gal4/+; P[B123]-RFP-moe/UAS-NcadRNAi_; (**h–h″**) _btl-Gal4/UAS-TrlRNAi; P[B123]-RFP-moe/+_. (**i–i″**) _btl-Gal4/+; P[B123]-RFP-moe/UAS-sdRNAi_. Source data for (**c, j**) are provided as a Source Data file.

permissive temperature of 29 °C for 48 h except for Fig. 5h, i (20 h). White pupae were collected for dissection and imaging.

**Live cell imaging**. HEK293T cells were transiently transfected with the plasmid using calcium phosphate transfection reagent. Cells were seeded in 35 mm glass bottom microwell (14 mm) dishes (MatTek Corporation). Transfection was performed when cells were cultured to ~50% confluence. For each transfection, ~200 ng of plasmid DNA was added. Images were obtained 24 h post-transfection by EVOS M7000 Imaging System (Thermo Fisher Scientific) with an environmental control unit incubation chamber maintained at 37 °C and 5% $CO_2$.

**EdU cell proliferation assay**. Pupae were dissected in cold PBS and the fat body and gut were gently removed. The samples were incubated in 1X EdU solution for 30 min at room temperature and then fixed with 4% formaldehyde in PBT at room temperature for 30 min. The samples were washed with PBS for three times and were permeabilized in PBS containing 1% Triton X-100 for 1 h. Subsequently, the samples were incubated in 5% goat serum in PBS and then were treated with Click-iT® reaction cocktail (Invitrogen, #C10337) at room temperature. After three times of wash, the samples were mounted in Vectashield. Statistics were performed using GraphPad Prism 7. Statistical significance values were calculated with unpaired two-tailed $t$-test.

**Ex vivo culturing of *Drosophila* trachea**. White pupae (0 h APF) were dissected in Grace's Insect culture medium with 5% fetal bovine serum (FBS; ThermoFisher/Invitrogen, 10270098) and Penicillin-Streptomycin (Sigma P4333, 100× stock solution) to impede microbial growth. The pupal trachea was incubated in Grace's medium supplemented with steroid hormone 20-hydroxyecdysone (20E)[42] and placed in depression slides. The specimens were imaged with a Zeiss Apotome microscope equipped with 10X objective.

**Immunohistochemistry**. Pupae were dissected in cold PBS and trachea were fixed in 4% formaldehyde. After several washes, the samples were permeabilized with 1% TritonX-100, blocked in 10% goat serum and followed by incubation with primary antibodies and secondary antibodies. Samples were mounted in Vectashield. Images were captured by an LSM Zeiss 900 inverted confocal laser scanning microscope. Primary antibodies: α-β-galactosidase (mouse, 1:100, Developmental Studies Hybridoma Bank, 1G9), α-Dlp (mouse, 1:100, Developmental Studies Hybridoma Bank, 13G8), α-Ncad (mouse, 1:100, Developmental Studies Hybridoma Bank, DN-EX), α-Hnt (mouse, 1:100, Developmental Studies Hybridoma Bank, 1G9), α-Rho1 (mouse, 1:100, Developmental Studies Hybridoma Bank, P1D9), α-MMP1 (mouse, 1:100, Developmental Studies Hybridoma Bank, 5H7B11/3A6B4/3B8D12), α-Serp (rabbit, 1:200, gift from Dr. Mark Krasnow), α-Verm (rabbit, 1:200, gift from Dr. Mark Krasnow), and α-Bnl antiserum (rabbit, 1:20 in M3, gift from Dr. Mark Krasnow). Secondary antibodies: α-mouse Alexa Fluor®488 (goat, 1:200, Jackson ImmunoResearch, 115-545-003), α-rabbit Alexa Fluor®488 (goat, 1:200, Jackson ImmunoResearch, 111-545-003), α-mouse Cyanine Cy™3 (goat, 1:200, Jackson ImmunoResearch, 115-165-003), α-rabbit Cyanine Cy™3 (goat, 1:200, Jackson ImmunoResearch, 111-165-003), α-mouse Alexa Fluor®647 (goat, 1:200, Jackson ImmunoResearch, 115-605-003), and α-rabbit Alexa Fluor®647 (goat, 1:200, Jackson ImmunoResearch, 111-605-003).

**Image quantification and statistical analysis**. To quantify ex-lacZ staining, the mean intensity of 555 nm fluorescence was measured in an area (containing approximately 10 tracheal progenitor cells). The value (with background fluorescence subtracted) was normalized with respect to GFP signal ($btl > GFP$). For each experiment, comparisons were made to control genotypes that were prepared and analyzed together with experimental genotypes in order to control for differences in staining and changes to laser intensity. Statistical analysis was done with GraphPad Prism or Excel. Statistical significance was calculated by $t$-test.

**Quantification of SPARK signal**. For quantitative analysis of the SPARK signal, images were processed in imageJ. The sum of droplets' pixel fluorescence intensity and the cells' pixel intensity were scored using Analyze Particle function in imageJ, as previously described[36].

$$\text{SPARK signal} = \frac{\Sigma \text{ pixel intensity of droplets}}{\Sigma \text{ pixel intensity of cells}}$$

**Measurement of progenitor migration**. The dorsal trunk was visualized by autofluorescence in the lumen. The length of tracheal metamere was identified by the distance between two neighboring TCs. The distance of progenitor migration is calculated as % of a metamere that progenitors proceed, as shown in Supplementary Fig. 12.

**Immunoprecipitation**. Protein extracts were prepared from larval trachea expressing Yki-FLAG or mutant form of Yki, YkiS168A:HA with RIPA buffer. 20 µl anti-Flag M2 magnetic beads or anti-HA agarose Affinity Gel antibody was

added to the lysate. The samples were incubated at 4 °C for 16 h under gentle agitation. The beads were washed for at least five times. Finally, the beads were eluted with 5 × SDS loading buffer. The eluted protein was analyzed by SDS/PAGE and followed by immunoblot analysis. Primary antibodies: α-Yki (rabbit, 1:100, gift from Dr. Jin Jiang), α-FLAG (mouse, 1:1000, Sigma, M2, #F1804) and α-HA (mouse, 1:500, ABclonal, #AE008). Secondary antibodies: HRP-conjugated α-mouse (goat, 1:5000, Jackson ImmunoResearch, #115-035-003) and HRP-conjugated α-rabbit (goat, 1:5000, Jackson ImmunoResearch, #111-035-144).

**In vitro kinase assay**. His-tagged Akt, PI3K, and AMPK proteins were purified from HEK293T cells. GST-tagged Yorkie proteins were purified from *E.coli* by glutathione agarose slurry and eluted with glutathione. The agarose was incubated with *E.coli* lysate for 16 h and then was washed with kinase washing buffer (40 mM Hepes and 200 mM NaCl, pH 7.5) for three times, and once with kinase assay buffer (40 mM Hepes, 50 mM KAC, and 5 mM MgCl2, pH 7.5). Purified kinase and Yki proteins were mixed with ATP or ATP-γ-S (500 µM) in kinase assay buffer. After 30 min kinase reaction at 30 °C, EDTA (final concentration 20 mM, pH 8.0) was added to terminate the reaction at 30 °C for 5 min. Then, PNBM (final concentration 2.5 mM) was added to form a thiophosphate ester side chain at 25 °C for 40 min. Immunoblotting was conducted using anti-Thiophosphate ester antibody (rabbit, 1:5000, Abcam, #ab92570) or phospho-specific antibody (rabbit, 1:1000, Cell Signaling Technology, #9920) to determine the kinase activity. See Key Resources Table and Reporting Summary for more detailed information of antibodies used in this assay. Secondary antibody: HRP-conjugated α-rabbit (goat, 1:5000, Jackson ImmunoResearch, #111-035-144).

**Chromatin immunoprecipitation**. The 1 h APF fly pupal trachea were dissected in cold PBS and were fixed in 1.8% formaldehyde at room temperature for 20 min. Brief mixing was conducted in between. The cross-linked chromatin was resuspended in RIPA buffer (140 mM NaCl, 10 mM Tris-HCl pH 8.0, 1 mM EDTA, 1% Triton X-100, 0.1% SDS, 0.1% sodium deoxycholate). The extracts were sonicated to produce DNA fragments with an average size of ~500 bp. 4 µg α-Yorkie antibodies (rabbit, 1:100, gift from Dr. Jin Jiang) was coupled to Dynabead protein G (Invitrogen, #10001D). Then, sonicated lysates were added and rotated overnight at 4 °C. Control IgG immunoprecipitations were performed in parallel. The chromatin samples were reverse cross-linked at 65 °C for 16 h.

Immunoprecipitated DNA was subjected to next generation sequencing using the Epicenter Nextera DNA Sample Preparation Kit or to real-time PCR. Library construction was performed using the High Molecular Weight tagmentation buffer, and tagmented DNA was amplified using 14 cycles of PCR. The libraries were then sequenced on a Novaseq according to manufacturer's standard protocols. The sequences were processed using *Fastqc*, and then low-quality bases and adaptor contamination were trimmed by *cutadapt*. Filtered reads were mapped to *Drosophila* genome using *BWA mem* algorithm. Peaks were called using *macs2 callpeak*[59]. Peaks were plotted using pyGenomeTracks. Motif analysis was performed by Homer. GO analysis of biological processes was conducted by PANTHER.

**Key resources table**.

| Reagent or Resource | Source | Identifier |
|---|---|---|
| Experimental Models: Organisms/Strains | | |
| D. melanogaster. btl-Gal4 | 60 | |
| D. melanogaster. btl-RFP-moe | 9 | |
| D. melanogaster. UAS-ykiRNAi | Vienna Drosophila RNAi Center | VDRC: 104523 |
| D. melanogaster. UAS-ykiRNAi | Vienna Drosophila RNAi Center | VDRC: 40497 |
| D. melanogaster. UAS-ykiRNAi | Tsinghua Stock Center | THU0579 |
| D. melanogaster. UAS-ykiRNAi | Bloomington Drosophila Stock Center | BDSC: 34067 |
| D.melanogaster.UAS-yki.S168A | Bloomington Drosophila Stock Center | BDSC: 28818 |
| D.melanogaster.UAS-CD8:GFP | Bloomington Drosophila Stock Center | BDSC: 5137 |
| D. melanogaster. UAS-YAP-SPARK | This paper | |
| D. melanogaster. Dl-Gal4 | 61 | |
| D. melanogaster. ex-lacZ | 45 | |
| D. melanogaster. UAS-InR$^{DN}$ | Bloomington Drosophila Stock Center | BDSC: 8252 |
| D. melanogaster. UAS-sdRNAi | | THU2534 |

| | | |
|---|---|---|
| | Tsinghua Stock Center | |
| *D. melanogaster. UAS-AktRNAi* | Tsinghua Stock Center | THU0552 |
| *D. melanogaster. UAS-Akt* | Bloomington Drosophila Stock Center | BDSC: 8191 |
| *D. melanogaster. UAS-wtsRNAi* | Vienna Drosophila RNAi Center | VDRC: 106174 |
| *D. melanogaster. UAS-dlpRNAi* | Bloomington Drosophila Stock Center | BDSC: 34091 |
| *D. melanogaster. UAS-bnlRNAi* | Bloomington Drosophila Stock Center | BDSC: 34572 |
| *D. melanogaster. UAS-hntRNAi* | [62] | |
| *D. melanogaster. UAS-dallyRNAi* | [63] | |
| *D. melanogaster. UAS-NcadRNAi* | Tsinghua Stock Center | THU2665 |
| *D. melanogaster. UAS-TrlRNAi* | Tsinghua Stock Center | THU03912.N |
| *D. melanogaster. UAS-AMPK^{DN}* | Bloomington Drosophila Stock Center | BDSC: 32112 |
| *D. melanogaster. yki-V5-Flag* | Vienna Drosophila RNAi Center | VDRC: 318237 |
| *D. melanogaster. UAS-Flag-yki* | [64] | |
| *D. melanogaster. UAS-ykiS168A:HA-GFP* | Bloomington Drosophila Stock Center | BDSC: 28816 |
| *D. melanogaster. dally:YFP* | Kyoto Stock Center | DGRC: 115511 |
| Cell line: | | |
| HEK293T | ATCC | CRL-11268 |
| Antibody/Kit | | |
| anti-β-galactosidase | Developmental Studies Hybridoma Bank | 40-1a |
| anti-Hnt | Developmental Studies Hybridoma Bank | 1G9 |
| anti-Dlp | Developmental Studies Hybridoma Bank | 13G8 |
| anti-Ncad | Developmental Studies Hybridoma Bank | DN-EX |
| anti-Serp | [65] | |
| anti-Verm | [65] | |
| anti-Rho1 | Developmental Studies Hybridoma Bank | P1D9 |
| anti-MMP1 | Developmental Studies Hybridoma Bank | 5H7B11/3A6B4/3B8D12 |
| anti-Thiophosphate ester | Abcam | #Ab92570 |
| Phospho-(Ser/Thr) Kinase Substrate Antibody Sampler Kit | Cell Signaling Technology | #9920 |
| anti-FLAG | Sigma | M2 |
| anti-HA | ABclonal | #AE008 |
| anti-Yki | [66] | |
| anti-Bnl | [67] | |
| HRP-conjugated α-mouse | Jackson ImmunoResearch | #115-035-003 |
| HRP-conjugated α-rabbit | Jackson ImmunoResearch | #111-035-144 |
| anti-mouse Alexa Fluor®488 | Jackson ImmunoResearch | #115-545-003 |
| anti-rabbit Alexa Fluor®488 | Jackson ImmunoResearch | #111-545-003 |
| anti-mouse Cyanine Cy™3 | Jackson ImmunoResearch | #115-165-003 |
| anti-rabbit Cyanine Cy™3 | Jackson ImmunoResearch | #111-165-003 |
| anti-mouse Alexa Fluor®647 | Jackson ImmunoResearch | #115-605-003 |
| anti-rabbit Alexa Fluor®647 | Jackson ImmunoResearch | #111-605-003 |
| ATP kit | Promega | #G7570 |
| Glucose kit | APLLYGEN | #E1011 |
| Click-iT® EdU Imaging Kits | Invitrogen | #C10337 |
| SMART-Seq v4 | Takara Bio | |
| RNeasy Micro Kit | Qiagen | #74004 |
| **Software and Algorithms** | | |
| ZEN (blue edition) | Carl Zeiss | N/A |
| ImageJ 1.53n | NIH | https://imagej.net |
| GraphPad Prism | GraphPad Software | https://www.graphpad.com/scientific-software/prism/ |
| Hisat2 | | https://ccb.jhu.edu/software/hisat2 |
| DESeq2 | Bioconductor | https://bioconductor.org/ |
| Homer 4.11 | | https://homer.ucsd.edu/homer/ |
| pyGenomeTracks 3.5.1 | | https://pygenometracks.readthedocs.io/ |
| ChIPseeker 1.22.1 | Bioconductor | https://bioconductor.org/ |
| PANTHER | | http://pantherdb.org/ |

**Reporting summary**. Further information on research design is available in the Nature Research Reporting Summary linked to this article.

## Data availability

The authors declare that all data supporting present study, including its supplementary information files, and the source data file, are available within this article and upon reasonable request from the corresponding author. The RNA sequencing data and ChIP-seq data generated and analyzed in this study have been deposited in the NCBI database under accession number GSE184856. The source data underlying Figures and Supplementary Figures are provided as a Source Data file.

## Code availability

All custom R scripts used to generate plots of RNA-seq or ChIP-seq are available at https://github.com/YueLi9104/Metabolic-control-of-tracheal-progenitors. The DOI for the Github repository is [https://doi.org/10.5281/zenodo.6474350]. Additional modified scripts are available upon request.

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

## Acknowledgements

We thank Drs Xiaohang Yang, Jin Jiang, Markus Affolter, Mark Krasnow, Stefan Lus-chnig, Hiroshi Nakato, Xianjue Ma, Richard Fehon, Howard Lipshitz, Edan Foley, John Reinitz for generously providing reagents; Core Facility of Drosophila Resource and Technology, SIBCB, CAS for injection service; Bloomington Drosophila Stock Center, Kyoto Stock Center, Vienna stock center, Tsinghua Stock Center for fly stocks; Developmental Studies Hybridoma Bank for antibodies; Feng Chen for technical advices; all members of Huang lab for discussions and constructive suggestions. This work has been financially supported by NSFC92168101, NSFC32070784 and Thousand Young Talent Program to H.H., NSFC32100670 and Postdoctoral Fellowship Foundation 2021M692825 to J.L.

## Author contributions

Conceptualization, H.H., Y.L; Biosensor, Q.Z., Y.Y., D.M.; Bioinformatics, formal analysis and software, Q.Y.Z., T.Y.G, T.F.L, F.N.X; Investigation and interpretation, Y.L., P.Z.D., H.H., H.L.L., J.L.L and all authors; Writing, T.B.K, J.M., H.H., Y.L.; Supervision, T.B.K., H.H., J.M.

## Competing interests

The authors declare no competing interests.
