## [Peer Review File · Nature Communications]

Metabolic control of progenitor cell propagation during
Drosophila tracheal remodelingREVIEWER COMMENTS

Reviewer #1 (Remarks to the Author):

In this study, Li and colleagues show that during metamorphosis, InR signaling activates tracheoblast proliferation and migration through modifying Yki activity. The authors show that metabolic activity in the trachea is depressed during pupation and identify metabolic genes and Hippo and Hippo pathway genes as those differentially expressed between larval and pupal datasets. In comparing fasting and normal fed L3 larval tracheoblast gene expression, InR/PI3K and Hippo pathways were among those showing the most significant changes. Using a InR pathway activity reporter, InR activity was seen to decline during puparium formation. Reporters of InR, Akt and AMP kinase activity were generated and used to follow activity in tracheoblasts. InR pathway activity was found to be suppressed upon metabolic depletion. A Hippo pathway reporter (expanded lacZ expression) was likewise examined and found to increase during the larval-pupal transition and under starvation conditions. Proliferation and migration were decreased by Yki RNAi, while Warts RNAi or Yki overexpression resulted in increased proliferation. Expression of a dominant negative insulin receptor inhibited Yki activity as assessed by the ex-lacZ reporter. A YAP/Yki reporter was generated and was also used to support this finding. Akt activity is regulated by InR and the authors found that Akt is able to phosphorylate and inactivate Yki in vitro. Authors show additional data consistent with an in vivo role of Akt in Yki phosphorylation. Authors identify candidate Yki target genes in tracheoblasts based on ChIP analysis, and show that yki RNAi downregulates expression of these candidates. RNAi analysis of yki regulated genes identify some that affect proliferation and others that affect migration.

All together, these results demonstrate an important interaction between InR and Hippo pathways that regulate tracheoblast behavior during metamorphosis. The data is largely compelling and the manuscript should be accepted with minor revisions. Authors should address the following:

Lines 93-5: The larval tracheal network consists of bilateral dorsal trunk (DT) tubes which are linked by dorsal branches (DBs), transverse connectives (TCs), and spiracular branches (SBs) in each of the 10 tracheal metameres (Tr1-Tr10; Fig. 1a,b, arrows) 31

> SB and TC do not link DTs...and a number of tracheal branches in the larval network are omitted (eg lateral trunk, visceral branch, ganglionic branch).

Lines 96-9: Clusters of tracheal progenitors are present in the 4th and 5th of the ten bilaterally symmetric SBs, and are visible in Fig. 1b by the fluorescence of red fluorescent protein (RFP) that was expressed by a transgene containing a promoter fragment that is specific for tracheal progenitors 32,33

> Ref 32 and 33 describe a btl enhancer element that is not specific for progenitors so far as reported in these refs, but is used by the authors to label the entire tracheal system; authors should clarify.

Lines 150-2: Phase separation and formation of droplets are achieved by the combination of multivalency and kinase activity-dependent (PPI).

> word for which PPI is an abbreviation (protein-protein interaction) is missing.

Lines 167-9: GFP droplets were abundant in L3 trachea of the three reporter lines, but not in animals with of reduced function of the respective kinases (Extended Data Fig. 3i-n).

> delete of from "with of reduced"

Lines 258- 67: Phosphorylation of FLAG-Yki was detected by an antibody that recognizes phospho-Akt substrates, but not by an antibody against phospho-AMPK substrates (Fig. 5d). Phosphorylation of Yki

by Akt was eliminated in the mutant YkiS168A (Fig. 5e). Second, expression of UAS-Akt increased the number of YAP-SPARK droplets, suggesting that phosphorylation of Yki is elevated by the up-regulation of Akt (Fig. 5f,g). Finally, in accordance with the inhibitory role of insulin in Yki signaling, overexpression of Akt in L3 animals increased phosphorylation of Yki, reduced EdU incorporation (Fig. 5h-j), and reduced progenitor migration (Fig. 5k-m). In sum, these observations indicate that Akt phosphorylates Yki at Ser168 and impedes Yki-dependent processes.

> The in vivo assays are indirect, consistent with Akt acting upstream of Yki, but not definitively showing that Akt phosphorylates Yki in vivo.

Lines 297-8: were identified by RNA-seq of trachea progenitors are also high-confident gene targets identified by the Yki ChIP-seq experiment (Fig. 6e).

> high confidence?

Reviewer #2 (Remarks to the Author):

This work by Li, Y. et al. reports that, during *Drosophila* metamorphosis, tracheal progenitor cells are activated through Yorkie-dependent proliferation and migration, by a metabolic deficit-induced signalling process. This process, the authors claim, is regulated by insulin and the Hippo pathway.

The authors take advantage of single-cell analyses to analyse the regulation of progenitor-cell activation during a developmental period that involves major metabolic changes. During this period, the beginning of *Drosophila* metamorphosis, the organism is subjected to a metabolic deficiency akin to starvation. The authors identify a regulatory mechanism through which metabolic depression drives progenitor cell activation and division, underlying organ remodelling.

Numerous studies point out that caloric restriction/metabolic depression, via the insulin pathway, has beneficiary effects on stem cell maintenance and tissue regeneration and making stem cells more likely to resist damage. Other studies have started discovering a growing web of interconnections between the Hippo pathway and metabolism. However, how the two pathways conjugate to allow for stem cell activation, and what consequences this has in the overall cell biology of stem cells, remains to be unraveled.

In this context, the finding that metabolic depression triggers stem cell activation in connection with the Hippo pathway is a noteworthy result, which will be of significance to the fields of organogenesis, growth, ageing and regeneration.

This work generated a great deal of transcriptomic data, and the questions are generally well addressed experimentally. The authors also have generated new tools which will be of significance to the field. However, I have major suggestions that will hopefully make some points clearer.

MAJOR COMMENTS

1) The figures are not very well exposed and sometimes are difficult to interpret. As a general suggestion for all figures, when one panel shows only one channel, it should be represented in gray scale and not in red or green. This makes analysis clearer. Colour panels should only be used when using colocalizations of more than one channel.

It is rather awkward that scale bars are only represented in one of the panels in groups of images. I understand that is because the scale is the same for all. However, in the following cases it is not clear:

Fig 2 h-j

Fig 3 a-c

2) Panels in Fig. 7 I' and I'' seem to be the same. This should be checked and changed to the appropriate panel.

3) In many places, the manuscript fails to clearly expose the hypothesis and conclusions of the experiments. Some sentences need to be revised and rewritten.

For instance:

- in lines 143-144 the authors say: "This suggests that insulin pathway activity decreases in these conditions of reduced metabolism" Reduced metabolism usually goes together with insulin pathway activity decreases. In addition, PI3K activity is stimulated by diverse growth factor receptors.
- in lines 175/176 the authors say: "These results suggest that insulin pathway is suppressed upon metabolic depletion." Is this really a conclusion drawn from these particular experiments? Metabolic depletion is generally a consequence of caloric restriction/insulin pathway suppression.

4) The methods section is not clear regarding some of the experiments. This should be revised. As an example, in Figure 3 the authors express Yki RNAi using a btl-GAL4 driver, but it is not clear if this was done under Gal80 conditions (the methods section does not refer to this particular RNAi under the Gal80ts section). If Gal80 was not used, does this mean YkiRNAi was expressed from embryonic stages?

Also XXX

5) The authors refer to the population of tracheoblasts examined throughout this study as stem cells. However, a more correct way should be to name them progenitor cells. This should be changed throughout the whole manuscript, title included.

6) In figure 2I" increased exlacZ expression is not clear in comparison with the L3 control. This should be quantified. These small differences could be attributed to different detection thresholds and needs an internal control. The same applies to figure 4 f-g.

7) Chen and Krasnow reported in 2014 that tracheal progenitors follow a stereotyped path out of the niche, tracking along a subset of tracheal branches destined for destruction. In this process the chemoattractant is Bnl, which is expressed dynamically ahead of the progenitors. In this work they show that Bnl knockdown abrogates progenitor outgrowth and migration. In this work the authors show that knockdown of Yki reduced progenitor migration. It is therefore essential to show if the Hippo pathway is interfering with the FGF pathway and consequently affecting migration.

8) Finally, for the sake of clarity, I suggest that the authors include a graphical model and refer to it in the discussion.

MINOR COMMENTS

- Line 163, "were" should be "was"
- Line 282, remove comma before brackets.
- Line 292, "factors" should be "factor".
- Line 299, "dependent" should be "depend".
- Line 354, "discover" should be "discovered".
- Line 442, "condition" should be "conditions".
- In the methods section, statistic methods are only briefly described and only for the EdU experiments.
- Text should be revised for clarity.

Reviewer #3 (Remarks to the Author):

The authors present an analysis of *yki* function in larval airway progenitors of the *Drosophila* respiratory system. They find that Insulin receptor (InR) signaling intersects with *yki* activation to control the proliferation of and migration of larval tracheoblasts. They further use ChIP experiments and RNA sequencing to define targets of *yki* at the relevant time points of pupal development and go on to validate some of the targets in the process of tracheoblast division and migration. Overall, the manuscript provides considerable novelty describing the regulation of tracheoblasts and presenting new reagents for in situ detection of InR, ATK and AMPK activation. I have some concerns on the coincidence of InR activation of the interpretation of the "cell migration" phenotypes, the analysis and presentation of the ChIP-seq experiment and the reliability of the SPARK-Yap reporter.

1) Presentation of reporters in the dorsal trunk cell instead of tracheoblasts. Lack of coincidence of GFP signals with *ex-lacZ* expression. Fig.2 shows convincingly that InR is active in the larval dorsal trunk cells and this activity is reduced in early pupae. The DT cells are destined to die (Chen and Krasnow 2014). The activity of the reporters is not shown in the tracheoblasts. How do the reporter dots differ in tracheoblasts, where *ex-lacZ* is increased? Fig 2L, 2LL" It should be possible to do double stainings for GFP and LacZ in dissected larval and early pupal trachea. On this context it would be helpful for the reader to explain how the differences in reporter dot number and size between panels 2I and 2I" are interpreted.

2) Cell migration phenotypes. The authors visualize groups of tracheoblasts with a cytoplasmic marker *moe-RFP* and measure how far the stainings extend on a metamere. Any intervention with the cytoskeleton (not necessarily migration) would extend or diminish (contract) the RFP staining. To state an effect on migration the authors would need to label and track single tracheoblast or do longer imaging (like in figure 1) to allow tracheoblast to extend into more posterior metameres. In figure 4 h,I, I have difficulties to see and interpret the differences in YAP-SPARK dots. *ex-lacZ* is clearly upon insulin addition is it possible to do a double staining for LacZ and GFP here? Figure 4j-q needs quantifications. Addition of 20E stimulates cell extensions but the effect of insulin on it is not visible to me. How was the quantification in 4m done? How many animals and how many metameres were counted? What does % of metamere mean? Fig5 K-I" and 5 m Similarly for 7J.

3) ChIP seq assay and presentation. How were the ChIP signals quantified relatively to the IgG control? The minimum would be to show the IgG tracks below the tracks in Fig6 using the same scales and describe how the data were normalized.

4) The specificity of the *Yki* reporter is questionable in vivo. Extended data figure 3 The dots are smaller in P compared to O but they are more in P compared to O. In other figures like Fig 4I more dots are taken to indicate stronger signaling. We need to see some type of reliable quantification.

REVIEWER COMMENTS

Reviewer #1 (Remarks to the Author):

In this study, Li and colleagues show that during metamorphosis, InR signaling activates tracheoblast proliferation and migration through modifying Yki activity. The authors show that metabolic activity in the trachea is depressed during puparation and identify metabolic genes and Hippo and Hippo pathway genes as those differentially expressed between larval and pupal datasets. In comparing fasting and normal fed L3 larval tracheoblast gene expression, InR/PI3K and Hippo pathways were among those showing the most significant changes. Using a InR pathway activity reporter, InR activity was seen to decline during puparium formation. Reporters of InR, Akt and AMP kinase activity were generated and used to follow activity in tracheoblasts. InR pathway activity was found to be suppressed upon metabolic depletion. A Hippo pathway reporter (expanded lacZ expression) was likewise examined and found to increase during the larval-pupal transition and under starvation conditions.

Proliferation and migration were decreased by Yki RNAi, while Warts RNAi or Yki overexpression resulted in increased proliferation. Expression of a dominant negative insulin receptor inhibited Yki activity as assessed by the ex-lacZ reporter. A YAP/Yki reporter was generated and was also used to support this finding. Akt activity is regulated by InR and the authors found that Akt is able to phosphorylate and inactivate Yki in vitro. Authors show additional data consistent with an in vivo role of Akt in Yki phosphorylation. Authors identify candidate Yki target genes in tracheoblasts based on ChIP analysis, and show that yki RNAi downregulates expression of these candidates. RNAi analysis of yki regulated genes identify some that affect proliferation and others that affect migration.

All together, these results demonstrate an important interaction between InR and Hippo pathways that regulate tracheoblast behavior during metamorphosis. The data is largely compelling and the manuscript should be accepted with minor revisions. Authors should address the following:

Authors' response: We thank this reviewer for acknowledging the significance and quality of our work.

Lines 93-5: The larval tracheal network consists of bilateral dorsal trunk (DT) tubes which are linked by dorsal branches (DBs), transverse connectives (TCs), and spiracular branches (SBs) in each of the 10 tracheal metameres (Tr1-Tr10; Fig. 1a,b, arrows) ³¹

> SB and TC do not link DTs...and a number of tracheal branches in the larval network are omitted (eg lateral trunk, visceral branch, ganglionic branch).

Authors' response: We thank this reviewer for pointing out this omission in our previous submission. We have revised Fig. 1a to include all the relevant branches as suggested by this reviewer.

Lines 96-9: Clusters of tracheal progenitors are present in the 4th and 5th of the ten bilaterally symmetric SBs, and are visible in Fig. 1b by the fluorescence of red fluorescent protein (RFP) that

was expressed by a transgene containing a promoter fragment that is specific for tracheal progenitors 32,33

> Ref 32 and 33 describe a *btl* enhancer element that is not specific for progenitors so far as reported in these refs, but is used by the authors to label the entire tracheal system; authors should clarify.

Authors' response: We thank this reviewer for pointing out this issue that was not adequately explained in our previous submission. In our study, we used a *P[B123]-RFP-moe* allele that harbors part of *btl* enhancer and marks activated progenitor cells ¹. This line was reported in the Chen and Krasnow paper. In our revised manuscript we cite this paper specifically when describing our experimental setup, and we apologize for our previous error.

Lines 150-2: Phase separation and formation of droplets are achieved by the combination of multivalency and kinase activity-dependent (PPI).

> word for which PPI is an abbreviation (protein-protein interaction) is missing.

Authors' response: We made the suggested change.

Lines 167-9: GFP droplets were abundant in L3 trachea of the three reporter lines, but not in animals with of reduced function of the respective kinases (Extended Data Fig. 3i-n).

> delete of from "with of reduced"

Authors' response: We made the suggested change.

Lines 258- 67: Phosphorylation of FLAG-Yki was detected by an antibody that recognizes phospho-Akt substrates, but not by an antibody against phospho-AMPK substrates (Fig. 5d). Phosphorylation of Yki by Akt was eliminated in the mutant YkiS168A (Fig. 5e). Second, expression of UAS-Akt increased the number of YAP-SPARK droplets, suggesting that phosphorylation of Yki is elevated by the up-regulation of Akt (Fig. 5f,g). Finally, in accordance with the inhibitory role of insulin in Yki signaling, overexpression of Akt in L3 animals increased phosphorylation of Yki, reduced EdU incorporation (Fig. 5h-j), and reduced progenitor migration (Fig. 5k-m). In sum, these observations indicate that Akt phosphorylates Yki at Ser168 and impedes Yki-dependent processes.

> The in vivo assays are indirect, consistent with Akt acting upstream of Yki, but not definitively showing that Akt phosphorylates Yki in vivo.

Authors' response: We agree with this reviewer and have revised the text accordingly (see page 11).

Lines 297-8: were identified by RNA-seq of trachea progenitors are also high-confident gene targets identified by the Yki ChIP-seq experiment (Fig. 6e).

> high confidence?

Authors' response: We made the suggested correction.

Reviewer #2 (Remarks to the Author):

This work by Li, Y. et al. reports that, during *Drosophila* metamorphosis, tracheal progenitor cells are activated through Yorkie-dependent proliferation and migration, by a metabolic deficit-induced signalling process. This process, the authors claim, is regulated by insulin and the Hippo pathway.

The authors take advantage of single-cell analyses to analyse the regulation of progenitor-cell activation during a developmental period that involves major metabolic changes. During this period, the beginning of *Drosophila* metamorphosis, the organism is subjected to a metabolic deficiency akin to starvation. The authors identify a regulatory mechanism through which metabolic depression drives progenitor cell activation and division, underlying organ remodelling.

Numerous studies point out that caloric restriction/metabolic depression, via the insulin pathway, has beneficiary effects on stem cell maintenance and tissue regeneration and making stem cells more likely to resist damage. Other studies have started discovering a growing web of interconnections between the Hippo pathway and metabolism. However, how the two pathways conjugate to allow for stem cell activation, and what consequences this has in the overall cell biology of stem cells, remains to be unraveled.

In this context, the finding that metabolic depression triggers stem cell activation in connection with the Hippo pathway is a noteworthy result, which will be of significance to the fields of organogenesis, growth, ageing and regeneration.

Authors' response: We thank this reviewer for his/her recognition of the significance of our work.

This work generated a great deal of transcriptomic data, and the questions are generally well addressed experimentally. The authors also have generated new tools which will be of significance to the field. However, I have major suggestions that will hopefully make some points clearer.

MAJOR COMMENTS

1) The figures are not very well exposed and sometimes are difficult to interpret. As a general suggestion for all figures, when one panel shows only one channel, it should be represented in gray scale and not in red or green. This makes analysis clearer. Colour panels should only be used when using colocalizations of more than one channel.

It is rather awkward that scale bars are only represented in one of the panels in groups of images. I understand that is because the scale is the same for all. However, in the following cases it is not clear:

Fig 2 h-j

Fig 3 a-c

Authors' response: As suggested by this reviewer, we changed one-channel images to grey images,

especially images showing SPARK reporters. We left images containing *btl>GFP* or *P[B123]-RFP-moe* in color for comparison in analysis.

We added scale bars for Fig. 2 and Fig. 3 as suggested by this reviewer.

2) Panels in Fig. 7 I' and I'' seem to be the same. This should be checked and changed to the appropriate panel.

Authors' response: We thank this reviewer for pointing this out and we have made appropriate replacements.

3) In many places, the manuscript fails to clearly expose the hypothesis and conclusions of the experiments. Some sentences need to be revised and rewritten.

For instance:

- in lines 143-144 the authors say: "This suggests that insulin pathway activity decreases in these conditions of reduced metabolism" Reduced metabolism usually goes together with insulin pathway activity decreases. In addition, PI3K activity is stimulated by diverse growth factor receptors.

- in lines 175/176 the authors say: "These results suggest that insulin pathway is suppressed upon metabolic depletion." Is this really a conclusion drawn from these particular experiments? Metabolic depletion is generally a consequence of caloric restriction/insulin pathway suppression.

Authors' response: We thank this reviewer for his/her suggestions. We have made significant changes to enhance the presentation of our work. Several sections were completely rewritten in the revised manuscript as highlighted in red.

4) The methods section is not clear regarding some of the experiments. This should be revised. As an example, in Figure 3 the authors express *Yki* RNAi using a *btl-GAL4* driver, but it is not clear if this was done under *Gal80* conditions (the methods section does not refer to this particular RNAi under the *Gal80ts* section). If *Gal80* was not used, does this mean *Yki*RNAi was expressed from embryonic stages?

Also XXX

Authors' response: The expression of *yki*RNAi is under the control of *tub-Gal80^{ts}*. We apologize for not adequately describing the experimental setup in our previous submission. We have rewritten the Methods section as suggested by the reviewer.

5) The authors refer to the population of tracheoblasts examined throughout this study as stem cells. However, a more correct way should be to name them progenitor cells. This should be changed throughout the whole manuscript, title included.

Authors' response: We revised it throughout the entire manuscript accordingly.

6) In figure 21'' increased *exlacZ* expression is not clear in comparison with the L3 control. This should be quantified. These small differences could be attributed to different detection thresholds and needs an internal control. The same applies to figure 4 f-g.

Authors' response: With regard to this reviewer's suggestion of internal control, we utilized *btl*-Gal4-driven *UAS-GFP* that serves as an internal control. The GFP signal is independent of these treatments. We performed quantitative analysis of ex-lacZ expression, as shown in Fig. 2p and Fig. 4j. The procedure is described in the Methods section.

7) Chen and Krasnow reported in 2014 that tracheal progenitors follow a stereotyped path out of the niche, tracking along a subset of tracheal branches destined for destruction. In this process the chemoattractant is Bnl, which is expressed dynamically ahead of the progenitors. In this work they show that Bnl knockdown abrogates progenitor outgrowth and migration.

In this work the authors show that knockdown of Yki reduced progenitor migration. It is therefore essential to show if the Hippo pathway is interfering with the FGF pathway and consequently affecting migration.

Authors' response: According to this reviewer's suggestion, we perturbed FGF pathway by expressing a dominant negative form of FGFR, *Btl^{DN}* in the trachea. Consistent with Chen and Krasnow, our results showed that perturbation of Btl abolished the migration of progenitors (Supplementary Fig. 11d,e and Supplementary Movie 4), which phenocopied knockdown of Yki. In addition, expression of *btl^{DN}* decreased activity of Yki in the trachea (See Supplementary Fig. 11a-c). These results provide further support an interplay between FGF pathway and Yki signaling in tracheal progenitor migration.

8) Finally, for the sake of clarity, I suggest that the authors include a graphical model and refer to it in the discussion.

Authors' response: According to this reviewer's suggestion, we added a signaling network diagram in the discussion (Supplementary Fig. 10).

MINOR COMMENTS

-Line 163, "were" should be "was"

-Line 282, remove comma before brackets.

-Line 292, "factors" should be "factor".

-Line 299, "dependent" should be "depend".

-Line 354, "discover" should be "discovered".

-Line 442, "condition" should be "conditions".

Authors' response: We made suggested changes. We thank the reviewer for the careful reading of our manuscript.

- In the methods section, statistic methods are only briefly described and only for the EdU experiments.

- Text should be revised for clarity.

Authors' response: According to this reviewer's suggestion, we added statistic and quantification methods for staining, SPARK signal, and progenitor migration in Methods section of our revised

manuscript.

Reviewer #3 (Remarks to the Author):

The authors present an analysis of *yki* function in larval airway progenitors of the *Drosophila* respiratory system. They find that Insulin receptor (InR) signaling intersects with *yki* activation to control the proliferation of and migration of larval tracheoblasts. They further use ChIP experiments and RNA sequencing to define targets of *yki* at the relevant time points of pupal development and go on to validate some of the targets in the process of tracheoblast division and migration. Overall, the manuscript provides considerable novelty describing the regulation of tracheoblasts and presenting new reagents for in situ detection of InR, AKT and AMPK activation. I have some concerns on the coincidence of InR activation of the interpretation of the “cell migration” phenotypes, the analysis and presentation of the ChIP-seq experiment and the reliability of the SPARK-Yap reporter.

Authors' response: We thank this reviewer for recognizing the novelty of our work.

1) Presentation of reporters in the dorsal trunk cell instead of tracheoblasts. Lack of coincidence of GFP signals with *ex-lacZ* expression. Fig.2 shows convincingly that InR is active in the larval dorsal trunk cells and this activity is reduced in early pupae. The DT cells are destined to die (Chen and Krasnow 2014). The activity of the reporters is not shown in the tracheoblasts. How do the reporter dots differ in tracheoblasts, where *exlacZ* is increased? Fig 2L, 2LL” It should be possible to do double stainings for GFP and LacZ in dissected larval and early pupal trachea.

On this context it would be helpful for the reader to explain how the differences in reporter dot number and size between panels 2I and 2I” are interpreted.

Authors' response: Regarding to the reviewer's suggestion of InR signal in tracheoblasts, we performed double staining of tracheoblasts for GFP and *lacZ*. The results showed that the signal of InR-SPARK reporter inversely correlated with the expression of *ex-lacZ* (Fig. 2n,o). Consistent with InR-SPARK in DT (Fig. 2i,i'), the number and size of GFP droplets in tracheoblast was reduced in white pupae, compared with L3 larvae. We calculated the ratio of pixel fluorescence intensity from cellular droplets relative to total pixel intensity of the cell, which represents the strength of SPARK signal (Fig. 2q). Detailed description is added in the text. The formula is provided in the Methods section.

2) Cell migration phenotypes. The authors visualize groups of tracheoblasts with a cytoplasmic marker *moe-RFP* and measure how far the stainings extend on a metamere. Any intervention with the cytoskeleton (not necessarily migration) would extend or diminish (contract) the RFP staining. To state an effect on migration the authors would need to label and track single tracheoblast or do longer imaging (like in figure 1) to allow tracheoblast to extend into more posterior metameres.

In figure 4 h,I, I have difficulties to see and interpret the differences in YAP-SPARK dots. *exlacZ* is clearly upon insulin addition is it possible to do a double staining for LacZ and GFP here? Figure 4j-q needs quantifications. Addition of 20E stimulates cell extensions but the effect of insulin on it

is not visible to me. How was the quantification in 4m done? How many animals and how many metameres were counted? What does % of metamere mean? Fig5 K-l” and 5 m Similarly for 7J.

Authors’ response: We agree with this reviewer that time-lapse imaging is favorable for analysis of migration. We provide steady images together with 5-hr movies (see Supplementary Movie 1 for Figure 1 and Supplementary Movie 3 for other Figures), as tracking single tracheoblast through intact cuticle is technically challenging.

We performed double staining of ex-lacZ and YAP-SPARK, according to this reviewer’s suggestion. It should be mentioned that the GFP droplets are liquid-like condensates and fixation impairs the fluorescent signal as well as the background fluorescence. The results showed that signal of YAP-SPARK was increased accompanying with the reduced expression of ex-lacZ upon the addition of insulin (Fig. 4h-k).

We provided quantification for Figure 5a-f corresponding to previous Fig. 4j-o, as shown in Fig. 5g. For previous Fig. 4p,q (current Fig. 5h,i), we identified 18 larvae showing precocious migration of tracheal progenitors in 20 animals expressing *InR^{DN}*.

For analysis of progenitor migration, we examined at a minimum 8 metameres in 8 animals for each condition. % of metamere represents migration distance of progenitors relative to the length of a metamere (see Supplementary Fig. 12). The velocity of migration is calculated as the % of tracheal metamere per min. The detailed methodology for measurement is shown in Supplementary Fig. 12 and Methods section. Statistics are shown in Source Data file.

We conducted this measurement for previous Fig. 5k-l”,m (current Fig. 6l-n) and Fig. 7j (current Fig. 8j) and quantification is shown in Fig. 6n and Fig. 8j.

3) ChIP seq assay and presentation. How were the ChIP signals quantified relatively to the IgG control? The minimum would be to show the igG tracks below the tracks in Fig6 using the same scales and describe how the data were normalized.

Authors’ response: We performed ChIP-seq with input for normalization (see input in Supplementary Fig. 8a). In parallel, we also conducted IgG control group. The beads with control IgG did not precipitate sufficient genomic DNA for library preparation. To validate our ChIP-seq results, we performed ChIP-qPCR for Yki targets identified by ChIP-seq. The results show that significant enrichment of Yki in binding regions of target gene compared with IgG control (Supplementary Fig. 8b).

4) The specificity of the Yki reporter is questionable in vivo. Extended data figure 3 The dots are smaller in P compared to O but they are more in P compared to O. In other figures like Fig 4I more dots are taken to indicate stronger signaling. We need to see some type of reliable quantification.

Authors’ response: We agree with this reviewer and apologize for not adequately documenting the quantification of SPARK signal in our previous submission. Throughout our work, we used procedures for quantifying SPARK signal in a manner that is identical to those described previously².

Since the phosphorylation of the substrate and/or kinase activity correlates with the size and abundance of GFP droplets, the ratio of pixel fluorescence intensity from cellular droplets relative to total pixel intensity of the cell represents the strength of SPARK signal. We added the formula in the Methods section and source data of SPARK quantification for previous Extended Data Figure 3 and Fig 4i is listed in Source Data file. The quantification of YAP-SPARK is shown in Fig. 4k and Fig. 6j.

$$\text{SPARK signal} = \frac{\Sigma \text{ pixel intensity of droplets}}{\Sigma \text{ pixel intensity of cells}}$$

- 1 Chen, F. & Krasnow, M. A. Progenitor outgrowth from the niche in *Drosophila trachea* is guided by FGF from decaying branches. *Science* **343**, 186-189, doi:10.1126/science.1241442 (2014).
- 2 Zhang, Q. *et al.* Visualizing Dynamics of Cell Signaling In Vivo with a Phase Separation-Based Kinase Reporter. *Mol Cell* **69**, 334-346 e334, doi:10.1016/j.molcel.2017.12.008 (2018).

REVIEWER COMMENTS

Reviewer #1 (Remarks to the Author):

I am satisfied with the responses and changes from the authors and support publication of the manuscript in its current form.

Reviewer #2 (Remarks to the Author):

In this revised version, Yue Li and colleagues have substantially improved their manuscript and provide new data sets.

The authors have discussed comments and questions submitted by the reviewers. They provided the requested experiments and changed the manuscript.

The manuscript has improved since the original submission.

However, the manuscript is still unclear on the link between the insulin pathway, Yki and the FGF pathway. The authors have limited to perform a similar experiment to the one published in Chen and Krasnow, 2014.

In the revised form of the manuscript the authors write:

"It appears that the Yki signaling in the progenitor interferes with FGF pathway. Perturbation of FGF pathway by expressing a dominant negative form of FGFR, BtlDN, reduced expression of ex-lacZ and phenocopied defective migration by Yki abrogation, suggesting that FGF may act upstream of Yki signaling (Fig. 3j-l, Supplementary Fig. 11)."

This is a very confusing interpretation of the results. As far as I can see in Sup Fig 11, btlDN expression decreases Yki activity in the trachea. Therefore, it is not Yki which interferes with the FGF pathway, but the FGF pathway which interferes with Yki activity and exlacZ expression.

The important question is to differentiate between Yki regulation of the FGF pathway from FGF regulation of the Hippo pathway. If Yki is upstream of FGF in progenitor cells, then all conclusions of the manuscript are acceptable. However, if it is the other way around, then the authors are not showing that signalling by the insulin pathway controls progenitor migration by regulating Yki, but are showing instead that the Insulin pathway regulates the FGF pathway and it is this pathway which is then regulating migration and Yki levels.

Is the FGF expression in decaying branches, proposed by Chen and Krasnow, induced by Yki? The authors should at least show if Yki and Bnl expression are lower in these branches and if Bnl expression and migration can be rescued by inducing Yki expression in decaying branches. I believe it is essential that the authors clarify this point.

Reviewer #3 (Remarks to the Author):

In the revised version the authors addressed all my concerns

REVIEWER COMMENTS

Reviewer #1 (Remarks to the Author):

I am satisfied with the responses and changes from the authors and support publication of the manuscript in its current form.

Reviewer #2 (Remarks to the Author):

In this revised version, Yue Li and colleagues have substantially improved their manuscript and provide new data sets.

The authors have discussed comments and questions submitted by the reviewers. They provided the requested experiments and changed the manuscript.

The manuscript has improved since the original submission.

However, the manuscript is still unclear on the link between the insulin pathway, Yki and the FGF pathway. The authors have limited to perform a similar experiment to the one published in Chen and Krasnow, 2014.

In the revised form of the manuscript the authors write:

"It appears that the Yki signaling in the progenitor interferes with FGF pathway. Perturbation of FGF pathway by expressing a dominant negative form of FGFR, BtlDN, reduced expression of *exlacZ* and phenocopied defective migration by Yki abrogation, suggesting that FGF may act upstream of Yki signaling (Fig. 3j-l, Supplementary Fig. 11)."

This is a very confusing interpretation of the results. As far as I can see in Sup Fig 11, *btlDN* expression decreases Yki activity in the trachea. Therefore, it is not Yki which interferes with the FGF pathway, but the FGF pathway which interferes with Yki activity and *exlacZ* expression.

The important question is to differentiate between Yki regulation of the FGF pathway from FGF regulation of the Hippo pathway. If Yki is upstream of FGF in progenitor cells, then all conclusions of the manuscript are acceptable. However, if it is the other way around, then the authors are not showing that signalling by the insulin pathway controls progenitor migration by regulating Yki, but are showing instead that the Insulin pathway regulates the FGF pathway and it is this pathway which is then regulating migration and Yki levels.

Is the FGF expression in decaying branches, proposed by Chen and Krasnow, induced by Yki? The authors should at least show if Yki and *Bnl* expression are lower in these branches and if *Bnl* expression and migration can be rescued by inducing Yki expression in decaying branches. I believe it is essential that the authors clarify this point.

Authors' response: We thank this reviewer for raising this point and we provide additional evidence to further clarify it. Indeed the expression of *fgf/bnl* in decaying branches was reduced upon RNAi targeting expression of *yki* and was increased by expression of constitutive active form of Yki (YkiS168A) or by overexpression of Yki (Supplementary Fig. 11f-j), supportive of the notion that

FGF expression is influenced by Yki. We also show that Yki is active in the Bnl-expressing branch, as assayed by *ex-lacZ* reporter (Supplementary Fig. 11a-e) and provide ChIP-seq data documenting occupancy of Yki in the promoter region of *bni* (Supplementary Fig. 11k). Perturbation of FGF signaling in the trachea prevented the migration of tracheal progenitors, a defect that phenocopied knockdown of *yki* and can be partially rescued by induced Yki expression (Supplementary Fig. 11l-n).

Reviewer #3 (Remarks to the Author):

In the revised version the authors addressed all my concerns

REVIEWERS' COMMENTS

Reviewer #2 (Remarks to the Author):

In this new revised form, the authors have addressed all my concerns and I am happy with the manuscript as it is.

REVIEWERS' COMMENTS

Reviewer #2 (Remarks to the Author):

In this new revised form, the authors have addressed all my concerns and I am happy with the manuscript as it is.

Authors' response: We complete the revision and thank reviewers for the constructive suggestions.